# Greatness in Simplicity: Unified Self-Cycle Consistency for Parser-Free Virtual Try-On

Chenghu Du[1]    Junyin Wang[1]    Shuqin Liu[4]    Shengwu Xiong[1,2,3,5,*]

[1]Wuhan University of Technology, [2]Shanghai AI Laboratory

[3]Sanya Science and Education Innovation Park, Wuhan University of Technology

[4]Wuhan Textile University, [5]Qiongtai Normal University

{duch, wjy199708, xiongsw}@whut.edu.cn, liushuqingwtu@foxmail.com

https://du-chenghu.github.io/USC-PFN/

## Abstract

Image-based virtual try-on tasks remain challenging, primarily due to inherent complexities associated with non-rigid garment deformation modeling and strong feature entanglement of clothing within human body. Recent groundbreaking formulations, such as in-painting, cycle consistency, and knowledge distillation, have facilitated self-supervised generation of try-on images. However, these paradigms necessitate the disentanglement of garment features within human body features through auxiliary tasks, such as leveraging 'teacher knowledge' and dual generators. The potential presence of irresponsible prior knowledge in the auxiliary task can serve as a significant bottleneck for the main generator ($e.g.$, 'student model') in the downstream task. Moreover, existing garment deformation methods lack the ability to perceive the correlation between the garment and the human body in the real world, leading to unrealistic alignment effects. To tackle these limitations, we present a new parser-free virtual try-on network based on unified self-cycle consistency (**USC-PFN**), which enables robust translation between different garments using just a single generator, faithfully replicating non-rigid geometric deformation of garments in real-life scenarios. Specifically, we first propose a self-cycle consistency architecture with a circular mode. It utilizes real unpaired garment-person images exclusively as input for training, effectively eliminating the impact of irresponsible prior knowledge at the model input end. Additionally, we formulate a Markov Random Field to simulate a more natural and realistic garment deformation. Furthermore, USC-PFN can leverage a general generator for self-supervised cycle training. Experiments demonstrate that our method achieves state-of-the-art performance on a popular virtual try-on benchmark.

## 1   Introduction

To provide consumers with an online try-on experience similar to physical stores, researchers have focused on image-based virtual try-on (VTON), allowing individuals to virtually try on garments available in online shops. The mainstream image-based approach of VTON involves the replacement of a person's clothing area with an in-shop garment image. However, all current datasets are paired, meaning one person corresponds to only one piece of clothing they are wearing. To capture underlying correlations between unpaired images from random combinations in the dataset, three architectures have been used: in-painting methods [8, 3, 9, 10, 4, 11–14], knowledge distillation methods [11, 6, 7], and cycle consistency methods[15, 2] (see Figure 1).

---

*Shengwu Xiong is corresponding author.

37th Conference on Neural Information Processing Systems (NeurIPS 2023).

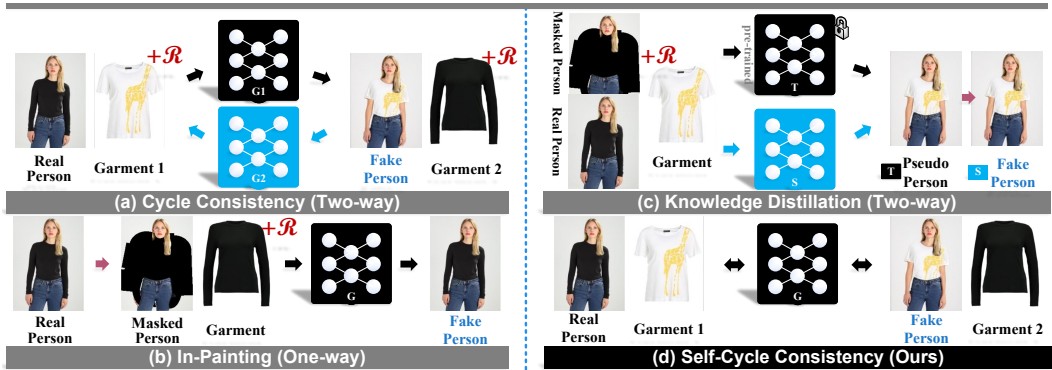

Figure 1: Comparison of virtual try-on pipelines. "$+\mathcal{R}$" denotes that additional person representation (*e.g.* semantic map) may be required. (a) cycle consistency approaches [1, 2] require two generators to implement bidirectional translation between various garment styles. (b) in-painting approaches [3–5] are implemented by reproducing the masked person's garment. (c) knowledge distillation approaches [6, 7] use pre-trained "teacher knowledge" to optimize student model. (d) USC-PFN utilizes self-cycle consistency without the need for additional person representation, enabling a single generator to perform bidirectional translation between various garment styles.

The in-painting-based method, inspired by [16], addresses variations in garment styles by masking the upper body of the person image and training the model to learn the reproduction of masked regions. This paradigm uses a person representation (*e.g.*, semantic map) as input to provide masked body information; however, slight errors in the person representation can result in occluded and unrealistic outcomes. Moreover, this paradigm faces challenges when dealing with cases where the same person tries on different garments due to their reliance on one-to-one paired garment-person images during training (see Figure 2). Therefore, the knowledge distillation-based structure [11] is proposed, which pretrains an in-painting-based teacher model to provide prior try-on result images for training the student model. This paradigm introduced a new pipeline that enables self-supervised training using only unpaired images as input. However, the potential presence of irresponsible teacher knowledge can still reproduce some issues of in-painting methods. Additionally, a method is proposed in [1, 2] that employs two CNNs to achieve the cycle training. In this approach, the output of $CNN_1$ as input is fed to $CNN_2$ and then minimizes the difference between the output of $CNN_2$ and the input of $CNN_1$ to achieve cycle consistency. However, the complex dual structure poses challenges in network convergence, and reliance on person representation throughout the entire network hinders the generation of high-quality virtual try-on results.

In this task, Thin Plate Splines (TPS) [17, 18, 8, 3, 10, 4, 11, 2, 12] is commonly used to deform garments. However, TPS focuses only on pixel transformations around the control point, often causing excessive distortion. Another non-rigid deformation technique, called Appearance Flow (AF) [19, 9, 6, 7, 14], predicts a pixel-level displacement field of garments for pixel-by-pixel transformations. However, it lacks spatial prior knowledge, resulting in an inaccurate perception of the spatial relationship between the garment and the human body. Recently, the Moving Least Squares (MLS) [20, 13] has shown promise in handling deformation for simple poses. However, there are difficulties in handling complex poses. These issues can lead to misalignment, hindering accurate simulation of the natural non-rigid deformation that occurs during the interaction between the garment and the body (see Figure 2).

Therefore, we propose a virtual parser-free try-on network based on self-cycle consistency to address the aforementioned limitations. First, we explore the relationship between garments and the human body. We use an auxiliary network to disentangle irrelevant features such as texture, color, and shape in the latent feature space of the human body. This approach preserves valuable features like depth and lighting while removing entanglements with unrelated features. Consequently, the network can authentically simulate the interaction between different garments and the human body in real-life scenarios, without being affected by the garment it wears. Furthermore, it has been confirmed that the person representation appearing at the input end of the network has potential negative impacts on performance [11, 6, 7]. Moreover, we argue that there is no inherent translation between multiple

domains in virtual try-on tasks. Therefore, we introduce a shared-weight circular architecture where the parser is shifted from the input end to the output end, specifically for loss computation. This approach allows us to bypass the parser during the inference stage. The main contributions of our paper can be summarized as follows:

• We propose USC-PFN, a novel parser-free virtual try-on network that generates highly realistic try-on results using only unpaired data as input and a single CNN as the generator, offering a new perspective for the virtual try-on task.

• We first leverage the Markov Random Field to estimate a deformation field to establish precise dense pixel-level correspondences between the in-shop garment and the person. This is a new approach to address the misalignment caused by a lack of spatial perception, and simulate the non-rigid deformation and interaction of the garment in real-life scenarios.

• We propose a concise and effective self-cycle consistency paradigm for the parser-free virtual try-on task, and extensive experiments on the popular benchmark show that our framework outperforms state-of-the-art methods both qualitatively and quantitatively.

## 2 Related Work

**Parser-Free Virtual Try-On** The formulations mentioned earlier (see Figure 1) can be subdivided based on whether the parser is used in image-to-image translation [21, 16]. It has been demonstrated that the parser can improve try-on results but may introduce unreliable semantic errors [11, 6, 7]. Initially, [8, 3] used pose heatmap and body mask as person representation to provide body's geometry information. Later, [10] introduced a parser to estimate an "after-try-on" semantic map instead of the above person representation. To avoid unreliable semantic information, [11, 6, 7] trained a "student model" without the parser using pretrained teacher knowledge. However, the potential presence of irresponsible teacher knowledge can still affect the "student model," making it challenging to train robust architectures without

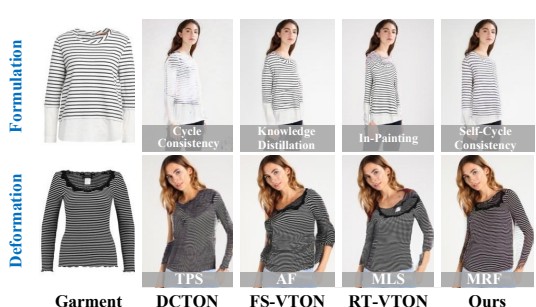

Figure 2: Comparisons of virtual try-on (DCTON [2], PF-AFN [6], FS-VTON [7], RT-VTON [13], and ours) and different deformation methods (TPS, AF, MLS, and our proposed MRF).

the parser in the input end. To address this challenge, we present a parser-free pipeline called self-cycle consistency. Our goal aims to achieve high-quality virtual try-on results using a general network, minimum input, and simple but efficient architecture.

**Non-rigid Deformation** In virtual try-on tasks, common non-rigid image deformation methods include Thin Plate Spline (TPS) [17, 18], Moving Least Squares (MLS) [20], and Appearance Flow (AF) [19]. As shown in Figure 2, TPS is the most widely used due to its ability to provide high-quality deformation while preserving the garment's image structure [8, 3, 10, 4, 11, 2, 12]. However, TPS is limited by its local mode, which only significantly affects pixels near control points. MLS is another method introduced recently, but it is not suitable for large-scale deformation [13]. AF is the most flexible method, deforming the garment by estimating pixel displacement vectors between the source and target shapes [9, 6, 7, 14]. However, none of these methods can fully mimic the natural interaction between the garment and the human body due to a lack of spatial perception. Our proposed method, which is based on Markov Random Field system with unpaired images as input, can obtain realistic spatial deformation.

**Self-Cycle Consistency** CycleGAN [1] utilized a self-supervised learning framework for cyclically learning the translation between unpaired images from different domains. [2] extended this with a deconstruction method for different garment styles to address the failure of cycle consistency in virtual try-on tasks. However, the complex dual-model structure and reliance on the parser hinder the generation of realistic results. To break through these limitations, following StarGAN [22], we propose a single-model self-cyclic learning translation paradigm between different garment styles.

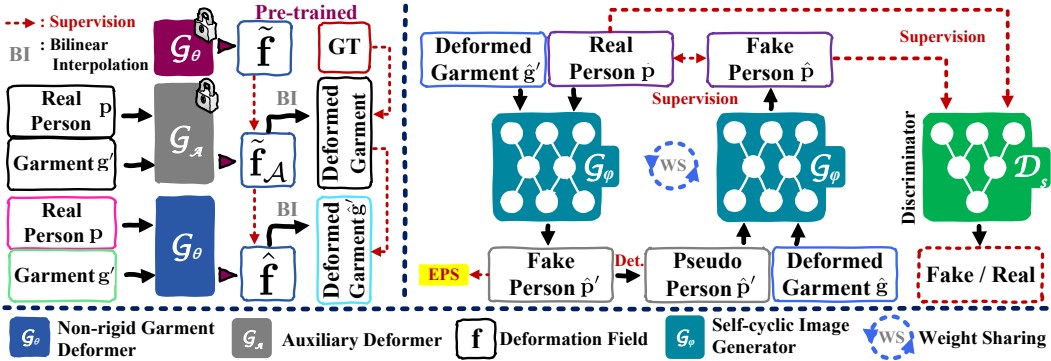

Figure 3: Overview of our self-cycle consistency network: paired garment-person images $(\mathbf{g}, \mathbf{p})$ and an arbitrary garment $\mathbf{g}'$ serve as the training data. **Left:** $\mathcal{G}_\theta$ is pre-trained using paired images to obtain the ground-truth deformation field $\tilde{\mathbf{f}}$ for the auxiliary deformer $\mathcal{G}_\mathcal{A}$ to optimize $\tilde{\mathbf{f}}_\mathcal{A}$. Then, $\mathcal{G}_\theta$ is further trained by using $\tilde{\mathbf{f}}_\mathcal{A}$ of $\mathcal{G}_\mathcal{A}^*$ with unpaired images. **Right:** $\mathcal{G}_\varphi$ takes unpaired $(\hat{\mathbf{g}}', \mathbf{p})$ to generate a fake person $\hat{\mathbf{p}}'$. Then, the real person $\mathbf{p}$ is reconstructed by re-feeding $\hat{\mathbf{p}}'$ and $\hat{\mathbf{g}}$ into $\mathcal{G}_\varphi$, ensuring self-cycle consistency $\mathbf{p} \approx \hat{\mathbf{p}}$. $\mathcal{D}_s$ is the discriminator that learns to distinguish real and fake images. **EPS** represents enhanced pixel-level supervision and **Det.** represents detaching tensor $\hat{\mathbf{p}}'$ from the computation graph.

## 3 Methodology

**Notation.** In this work, we use $(\mathbf{g}, \mathbf{p})$ to represent **paired** garment-person images where the person $\mathbf{p}$ is wearing garment $\mathbf{g}$. Moreover, we randomly select the garment image $\mathbf{g}'$ to form **unpaired** garment-person images $(\mathbf{g}', \mathbf{p})$ with $\mathbf{p}$. The try-on result of $(\mathbf{g}, \mathbf{p})$ and $(\mathbf{g}', \mathbf{p})$ are denoted as $\hat{\mathbf{p}}$ and $\hat{\mathbf{p}}'$, respectively. In general, "′" represents the target image, and "ˆ" denotes the transformation result. In addition, "↦" represents the mapping relationship of the entire architecture, while "⇒" represents the mapping relationship of the sub-modules.

**Overview.** The objective of the virtual try-on task is to naturally dress an arbitrary in-shop garment image $\mathbf{g}' \in \mathbb{R}^{3 \times H \times W}$ on a reference person image $\mathbf{p} \in \mathbb{R}^{3 \times H \times W}$ to obtain a try-on result $\hat{\mathbf{p}}' \in \mathbb{R}^{3 \times H \times W}$. As illustrated in Figure 3, we first propose a self-cycle consistency framework, USC-PFN, a new virtual try-on paradigm that does not require the human parser and other person representation at the model input end for generating highly realistic try-on images. The framework is broken up into two sub-networks: (i) a **non-rigid garment deformer** $\mathcal{G}_\theta$, and (ii) a **self-cyclic image generator** $\mathcal{G}_\varphi$. They are mainly used to construct a cyclic mapping $V_t : (\mathbf{p}, \mathbf{g}') \mapsto \hat{\mathbf{p}}', (\hat{\mathbf{p}}', \mathbf{g}) \mapsto \hat{\mathbf{p}}$, where $\mathbf{p} \approx \hat{\mathbf{p}}$, *i.e.*, $\hat{\mathbf{p}}$ is the same to the real image $\mathbf{p}$.

### 3.1 Formulation

**Garments Deformation.** In $(\mathbf{p}, \mathbf{g}') \mapsto \hat{\mathbf{p}}'$, the initially flat garment $\mathbf{g}'$ must be deformed to match the posture of the person $\mathbf{p}$ before synthesizing the target try-on result $\hat{\mathbf{p}}'$, denotes as $(\mathbf{p}, \mathbf{g}') \overset{\theta}{\Rightarrow} \hat{\mathbf{g}}', (\mathbf{p}, \hat{\mathbf{g}}') \overset{\varphi}{\Rightarrow} \hat{\mathbf{p}}'$. Our key idea involves using $\mathcal{G}_\theta$ to model the non-rigid deformation based on Markov Random Field (MRF) [23], to seek an optimal spatial transformation solution $\hat{\mathbf{f}}^* \in \mathbb{R}^{2 \times H \times W}$ (*i.e.* optimal deformation field $\hat{\mathbf{f}}^* = \mathcal{G}_\theta^*(\mathbf{p}, \mathbf{g}')$) that minimizes the dissimilarity between the warped garment $\hat{\mathbf{g}}'$ and the garment $\hat{\mathbf{p}}'_\mathbf{g}$ worn on the target result $\hat{\mathbf{p}}'$, where $\hat{\mathbf{g}}'$ is obtained by using bilinear interpolation $(\delta)$: $\hat{\mathbf{g}}' = \varrho(\hat{\mathbf{f}}, \mathbf{g}')$. This can be formalized as minimizing the MRF energy function:

$$\text{minimize} \sum_{<\tilde{\mathbf{f}}_\mathcal{A}, \hat{\mathbf{p}}'_\mathbf{g}> \in \mathcal{V}^+} \left\{ E_{dat}\left(\tilde{\mathbf{f}}_\mathcal{A}, \hat{\mathbf{f}}\right) + \lambda_r E_{reg}\left(\hat{\mathbf{f}}\right) + \lambda_p E_{pix}\left(\hat{\mathbf{p}}'_\mathbf{g}, \hat{\mathbf{g}}'\right) \right\}, \tag{1}$$

where $\lambda_r$ and $\lambda_p$ are trade-off hyper-parameters. $\mathcal{V}^+$ represents the prior pseudo-label from the pre-trained garment deformer and $\tilde{\mathbf{f}}_\mathcal{A}$ is the deformation field of $\hat{\mathbf{p}}'_\mathbf{g}$.

$$E_{dat} = \text{argmin}_{\hat{\mathbf{f}}} \left\| \tilde{\mathbf{f}}_\mathcal{A} - \hat{\mathbf{f}} \right\|, \quad E_{reg} = \mathbf{w}_\phi R_{sm}\left(\hat{\mathbf{f}}\right), \quad E_{pix} = \text{argmin}_{\hat{\mathbf{f}}} D\left[\hat{\mathbf{p}}'_\mathbf{g}, \hat{\mathbf{g}}'\right], \tag{2}$$

where $E_{reg}$ is regularization term of $\hat{\mathbf{f}}$, which composed of spatial position weights $\mathbf{w}_\phi$ of pixel points and a smoothing term $R_{sm}$. $R_{sm}$ is used to ensure that each element in $\hat{\mathbf{f}}$ and its neighborhood have similar values. $D\left[\cdot,\cdot\right]$ indicates quantification of similarity or difference. During garment deformation, we find that the deformation range of the edge region is often larger than the central region. Therefore, we use the two-dimensional Gaussian distribution [24] with the centroid and variance of pixel coordinates in the cloth region, as $\mathbf{w}_\phi$ to maintain the flexibility of the edge region.

In $(\hat{\mathbf{p}}', \mathbf{g}) \mapsto \hat{\mathbf{p}}$, this process is similar to the one above, *i.e.*, $(\hat{\mathbf{p}}', \mathbf{g}) \overset{\theta}{\Rightarrow} \hat{\mathbf{g}}, (\hat{\mathbf{p}}', \hat{\mathbf{g}}) \overset{\varphi}{\Rightarrow} \hat{\mathbf{p}}$, but the difference is that we remove $E_{dat}$ from Eq.(1) to ensure more accurate results.

**Synthesis of Try-On Results.** Recall that the goal of the virtual try-on task is to synthesize the target try-on result after warping the garment as if it was shot from realistic scenes. Therefore, reconstructing the real image $\mathbf{p}$ is an essential task to form self-supervised training, which can be formalized as minimizing the following objective:

$$\text{minimize} \sum_{<\hat{\mathbf{g}},\mathbf{p}>\in\mathcal{V}} \text{argmin}_{\hat{\mathbf{p}}} \, D\left[\mathbf{p}, \hat{\mathbf{p}}\right], \text{ with } (\hat{\mathbf{p}}', \hat{\mathbf{g}}) \overset{\varphi}{\Rightarrow} \hat{\mathbf{p}}, \tag{3}$$

where $\mathcal{V}$ is training set. $\hat{\mathbf{p}}'$ denotes $\mathbf{p}$ wearing the garment $\mathbf{g}'$. This differentiable generation appears to have completeness, however, the absence of $\hat{\mathbf{p}}'$ in $\mathcal{V}$ forces the process to be compensated by extra person representation $\mathcal{R}$ (semantic maps, pose heatmaps, etc.) as input. This means that results are heavily constrained by $\mathcal{R}$, and any errors that occur can directly affect the final try-on effect. To tackle this problem, [6, 7] utilized a pre-trained and parser-based "teacher network" $\mathcal{T}$ to generate $\hat{\mathbf{p}}'$:

$$\hat{\mathbf{p}}' = \mathcal{T}^*\left(\mathcal{R}, \mathbf{p}, \hat{\mathbf{g}}'\right), \quad \hat{\mathbf{p}} = \mathcal{S}\left(\hat{\mathbf{p}}', \hat{\mathbf{g}}\right). \tag{4}$$

Yet they cannot handle that irresponsible teacher knowledge brings to the learning process of their "student network" $\mathcal{S}$. [2] introduced the cycle-consistency structure that employed and trained cyclically two CNNs (*e.g.*, $\mathcal{N}_1$ and $\mathcal{N}_2$), where let $\mathcal{N}_1$ provide $\hat{\mathbf{p}}'$ for $\mathcal{N}_2$:

$$\hat{\mathbf{p}}' = \mathcal{N}_1\left(\mathcal{R}, \mathbf{p}, \hat{\mathbf{g}}'\right), \quad \hat{\mathbf{p}} = \mathcal{N}_2\left(\mathcal{R}, \hat{\mathbf{p}}', \hat{\mathbf{g}}\right), \tag{5}$$

where the structural difference between $\hat{\mathbf{p}}$ and $\mathbf{p}$ is minimized to ensure cycle-consistency. However, decomposing the body and then synthesizing it back together based on semantics leads to occlusion and artifacts due to some parsing errors. Relying on prior knowledge weakens the learning of the garment of adaptive spatial structure on the human body. Additionally, the dual-model structure is complex, resulting in longer training times and unstable convergence. Inspired by [22], we reformulate the synthesis process as an unpaired garment style translation problem. Specifically, we provide $\hat{\mathbf{p}}'$ for ourselves ($\mathcal{G}_\varphi$) by ourselves (weight-sharing structure):

$$\hat{\mathbf{p}} = \mathcal{G}_\varphi\left(\hat{\mathbf{p}}', \hat{\mathbf{g}}\right), \text{ with } \hat{\mathbf{p}}' = \mathcal{G}_\varphi\left(\mathbf{p}, \hat{\mathbf{g}}'\right). \tag{6}$$

**Inference.** During inference, our goal is to synthesize the desired try-on result $\hat{\mathbf{p}}$ just by feeding arbitrary garment-person pairs $(\mathbf{p}, \mathbf{g}')$ you need into $\mathcal{G}_\theta^*$ and $\mathcal{G}_\varphi^*$:

$$\hat{\mathbf{p}} = \mathcal{G}_\varphi^*\left(\mathbf{p}, \hat{\mathbf{g}}'\right), \text{ with } \hat{\mathbf{g}}' = \varrho\left(\mathcal{G}_\theta^*\left(\mathbf{p}, \mathbf{g}'\right), \mathbf{g}'\right). \tag{7}$$

### 3.2 Network Architecture

**Non-rigid Garment Deformer (NGD) $\mathcal{G}_\theta$.** The deformer $\mathcal{G}_\theta$ can adopt arbitrary encoder-decoder networks. As shown in Eq. (1), to fit Markov Random Field, we first pre-trained $\mathcal{G}_\theta$ by feeding paired images $(\mathbf{p}, \mathbf{g})$ to learn a real deformation field $\tilde{\mathbf{f}}$, *i.e.* $\tilde{\mathbf{f}} = \mathcal{G}_\theta\left(\mathbf{p}, \mathbf{g}\right)$. At this point, pre-trained $\mathcal{G}_\theta$ contains the full latent alignment information, including shape, color, perspective changes, texture variations, shadows and lighting, and depth of field, between the paired images $(\mathbf{p}, \mathbf{g})$. However, features such as color, shape, and texture do not need to be learned, because the input garment varies greatly during inference, but $\mathbf{g}$ and $\mathbf{p}$ used for training are paired. Therefore, $\mathcal{G}_\theta$ is necessarily invalid when unpaired images are fed. To disentangle the aforementioned feature entanglement, we employ an auxiliary deformer $\mathcal{G}_\mathcal{A}$, which takes the garment $\mathbf{g}$, prior Densepose descriptor $\mathbf{p_d} \in \mathbb{R}^{1\times H\times W}$ [25], and the pose heatmap $\mathbf{p_h} \in \mathbb{R}^{18\times H\times W}$ as input to learn the same goal as pre-trained $\mathcal{G}_\theta$, *i.e.*, $\tilde{\mathbf{f}}_\mathcal{A} = \mathcal{G}_\mathcal{A}\left(\mathbf{p_d}, \mathbf{p_h}, \mathbf{g}\right)$. Because $\mathbf{p_d}$ and $\mathbf{p_h}$ are not correlated with $g$ in terms of color, texture, and shape, $\mathcal{G}_\mathcal{A}$ does not face the challenges mentioned above and can handle any type of garment effortlessly. However, unfortunately, $\mathbf{p_d}$ and $\mathbf{p_h}$ also lack perspective changes, shadows, lighting,

and depth of field. As a result, the deformation results obtained are often not realistic. Therefore, we use $\tilde{\mathbf{f}}$ to supervise the training of $\mathcal{G}_\mathcal{A}$, $\tilde{\mathbf{f}}_\mathcal{A} \approx \tilde{\mathbf{f}}$, supplementing its spatial perception abilities. After obtaining the optimal $\mathcal{G}_\mathcal{A}^*$, we continue to train the deformer $\mathcal{G}_\theta$ by taking unpaired $(\mathbf{p}, \mathbf{g}')$ as input and $\mathbf{f}_\mathcal{A}$ as supervision, to disentangle the feature entanglement of $\mathcal{G}_\theta$:

$$\tilde{\mathbf{f}}_\mathcal{A} \approx \hat{\mathbf{f}}, \;\; \text{with} \;\; \hat{\mathbf{f}} = \mathcal{G}_\theta\left(\mathbf{p}, \mathbf{g}'\right), \;\; \tilde{\mathbf{f}}_\mathcal{A} = \mathcal{G}_\mathcal{A}^*\left(\mathbf{p_d}, \mathbf{p_h}, \mathbf{g}'\right). \tag{8}$$

**Self-cyclic Image Generator (SIG) $\mathcal{G}_\varphi$.** The generator $\mathcal{G}_\varphi$ can adopt arbitrary encoder-decoder networks. Our goal is to optimize $\mathcal{G}_\varphi$ by taking unpaired images $(\mathbf{p}, \hat{\mathbf{g}}')$. The training process is divided into two steps: 1) generating the try-on result $\hat{\mathbf{p}}'$ by taking unpaired images $(\mathbf{p}, \hat{\mathbf{g}}')$ as input, *i.e.* $\hat{\mathbf{p}}' = \mathcal{G}_\varphi\left(\mathbf{p}, \hat{\mathbf{g}}'\right)$. Immediately after, 2) taking $\hat{\mathbf{p}}'$ and warped garment $\hat{\mathbf{g}}$ as input of $\mathcal{G}_\varphi$ to generate try-on result $\hat{\mathbf{p}}$, *i.e.* $\hat{\mathbf{p}} = \mathcal{G}_\varphi\left(\hat{\mathbf{p}}', \hat{\mathbf{g}}\right)$. Different from the current methods that rely on an additional network to provide prior pseudo-label $\hat{\mathbf{p}}'$, training $\mathcal{G}_\varphi$ directly with self-cycle consistency is not work due to the absence of ground truth of $\hat{\mathbf{p}}'$, $\mathcal{G}_\varphi$ is more inclined to learn the mapping: $\mathbf{p} \mapsto \hat{\mathbf{p}}$ when we jointly perform the two steps. Therefore, we introduce additional supervision on the upper body of the generated $\hat{\mathbf{p}}'$ to take over the role of the label in [22]. In simple terms, the functionality of the person representation $\mathcal{R}$ has shifted from the input side to the supervision side. In addition, the self-cycle consistency can be ensured by enforcing $\hat{\mathbf{p}} \approx \mathbf{p}$ to establish self-supervision in step 2.

## 3.3 Self-Cycle Consistency Training Strategy

**Training of NGD.** As shown in Eq.(1), Eq.(2), and Eq.(8), to achieve the MRF energy minimization, we design multiple loss functions. $E_{dat}$ is implemented with the pixel-level $\mathcal{L}_1$ loss to encourage the similarity between $\hat{\mathbf{f}}$ and $\tilde{\mathbf{f}}_\mathcal{A}$. $E_{reg}$ is composed of the weights $\mathbf{w}_\phi$ and the smoothing term $R_{sm}$ (implemented as second-order smooth constraint $\mathcal{L}_{sec}$ [6]). $E_{pix}$ is implemented with $\mathcal{L}_1$ loss and VGG perceptual loss $\mathcal{L}_{per}$ [26] as the evaluation of structural similarity. Finally, we define the overall loss function of NGD as:

$$\mathcal{L}_{ngd} = \mathcal{L}_1^{dat} + \lambda_r \mathcal{L}_{sec} + \lambda_p\left(\mathcal{L}_1 + \mathcal{L}_{per}\right), \tag{9}$$

where $\lambda_r$ and $\lambda_d$ are hyperparameters that balances the relative contributions of sub-loss terms.

**Training of SIG.** To self-supervised cyclic optimization of SIG, we propose a self-cycle consistency loss consisting of $\mathcal{L}_1$ loss and perceptual loss $\mathcal{L}_{per}$, respectively, to maintain consistency between the try-on result $\hat{\mathbf{p}}$ and the real person $\mathbf{p}$:

$$\mathcal{L}_{scyc} = \|\mathbf{p} - \hat{\mathbf{p}}\|_1 + \|\mathbf{p} - \hat{\mathbf{p}}\|_{per}. \tag{10}$$

To make $\hat{\mathbf{p}}'$ as close as possible to the real distribution of person, we employ a discriminator $\mathcal{D}_s$ to introduce adversarial loss:

$$\mathcal{L}_{adv}^D = \mathbb{E}_{\mathbf{p}}\left[\log\left(\mathcal{D}_s\left(\mathbf{p}\right)\right)\right] + \mathbb{E}_{\hat{\mathbf{g}}',\mathbf{p}}\left[\log\left(1 - \mathcal{D}_s\left(\mathcal{G}_\varphi\left(\hat{\mathbf{g}}', \mathbf{p}\right)\right)\right)\right]. \tag{11}$$

Additionally, the adversarial loss $\mathcal{L}_{adv}^G$ of generator $\mathcal{G}_\varphi$ is introduced simultaneously. In addition to the aforementioned losses, we propose a skin reconstruction loss $\mathcal{L}_{sr}$ to forcefully supervise the skin reconstruction of $\hat{\mathbf{p}}'$ in the arm and neck regions. To do this, we pre-trained a model SR, which can be fed into deformed garment $\hat{\mathbf{g}}'$ and parts of the skin $\mathbf{s_p}$ to reconstruct the envisaged arm and neck regions of $\hat{\mathbf{p}}'$:

$$\mathcal{L}_{sr} = \|\text{SR}\left(\hat{\mathbf{g}}', \mathbf{s_p}\right) - \hat{\mathbf{p}}_\mathbf{s}'\|_1, \tag{12}$$

where $\hat{\mathbf{p}}_\mathbf{s}'$ denotes skin regions of $\hat{\mathbf{p}}'$. Furthermore, to generate the garment part of $\hat{\mathbf{p}}'$ and preserve the invariant body content (*e.g.* head, trouser), we introduce the garment reconstruction loss $\mathcal{L}_{gr}$ and content preservation loss $\mathcal{L}_{cp}$:

$$\mathcal{L}_{gr} = \left\|\hat{\mathbf{p}}_\mathbf{g}' - \hat{\mathbf{g}}'\right\|_1, \;\; \mathcal{L}_{cp} = \left\|\hat{\mathbf{p}}_\mathbf{c}' - \mathbf{p_c}\right\|_1, \tag{13}$$

where $\hat{\mathbf{p}}_\mathbf{g}'$ denotes garment part of $\hat{\mathbf{p}}'$, $\hat{\mathbf{p}}_\mathbf{c}'$ and $\mathbf{p_c}$ denote the same content of $\hat{\mathbf{p}}'$ and $\mathbf{p}$, respectively.[2] Finally, we define the overall loss function of SIG as:

$$\mathcal{L}_{sig}^t = \lambda_{scyc}\mathcal{L}_{scyc} + \lambda_{adv}^G\mathcal{L}_{adv}^G + \lambda_{sr}\mathcal{L}_{sr} + \lambda_{gr}\mathcal{L}_{gr} + \lambda_{cp}\mathcal{L}_{cp}. \tag{14}$$

To prevent the unreliable result $\hat{\mathbf{p}}'$ generated in **step 1** to wrongly guide **step 2**, we introduce an adjustable gated mechanism late in the training to prevent irresponsible backpropagation:

$$\mathcal{L}_{sig} = \boldsymbol{\alpha}\mathcal{L}_{sig}^t, \;\; \text{with} \;\; \boldsymbol{\alpha} = \begin{cases} 1, & \|\mathbf{p} - \hat{\mathbf{p}}'\|_1 > \|\mathbf{p} - \hat{\mathbf{p}}\|_1, \\ 0, & \text{otherwise}, \end{cases} \tag{15}$$

where $\boldsymbol{\alpha}$ is the adjustable gated factor that determines whether backpropagation is enabled.

---

[2]We provide the means to obtain $\mathbf{s_p}$, $\hat{\mathbf{p}}_\mathbf{s}'$, $\hat{\mathbf{p}}_\mathbf{g}'$, and $\hat{\mathbf{p}}_\mathbf{c}'$ in the supplementary material.

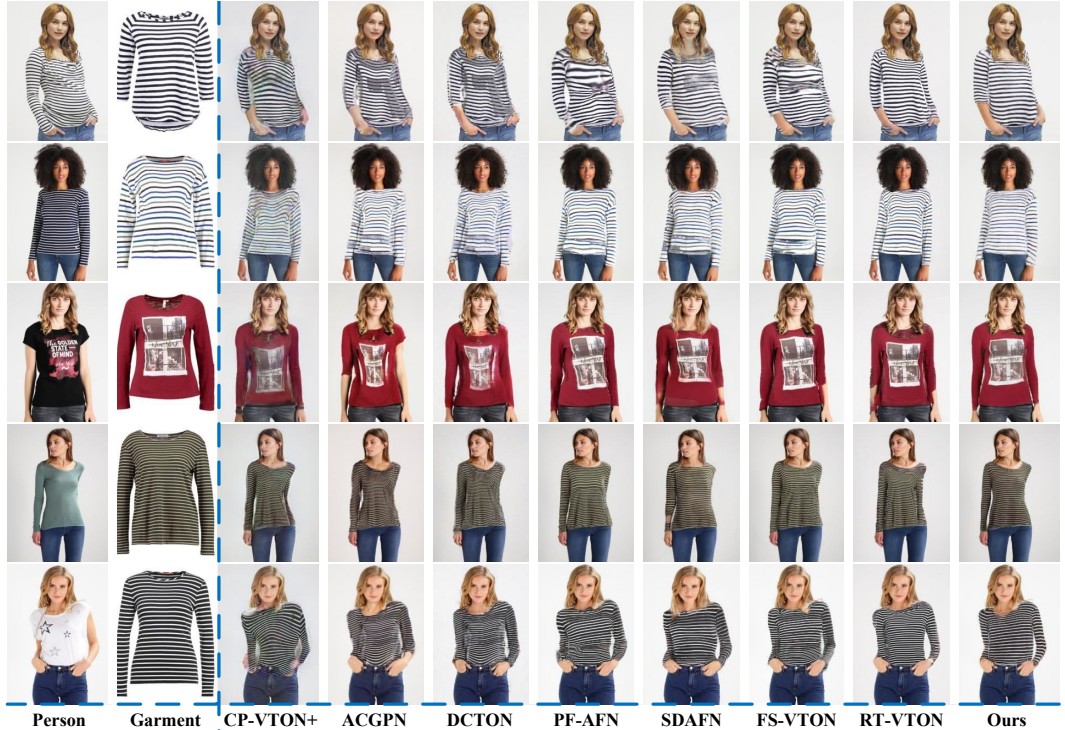

| Person | Garment | CP-VTON+ | ACGPN | DCTON | PF-AFN | SDAFN | FS-VTON | RT-VTON | Ours |

Figure 4: Qualitative results of different methods (CP-VTON+[27], ACGPN[4], DCTON[2], PF-AFN[6], SDAFN[28], RT-VTON[13], FS-VTON[7], and ours) in the unpaired setting.

## 4 Experiments

**Dataset.** We conduct experiments using the VITON dataset [8], which consists of 16,253 image groups. Each group includes a front-view female image $\mathbf{p}$, an in-shop garment image $\mathbf{g}$ with its mask $\mathbf{p_M}$, a reference semantic map $\mathbf{p_p}$, and a pose heatmap $\mathbf{p_h}$. The image size for each is 256 pixels $\times$ 192 pixels. The training set contains 14,221 groups, and the test set contains 2,032 groups. All evaluations and visualizations are performed using the test set. Note that the in-shop garment and the garment worn on the person are identical in the training set, while they differ in the test set.

**Implementation Details.** The USC-PFN is implemented in PyTorch and trained on a single Nvidia Tesla V100 GPU running Ubuntu 16.04. During training, a batch size of 16 is used for 100 epochs, and the Adam optimizer [29] is employed with parameters $\beta_1 = 0.5$ and $\beta_2 = 0.999$, and the initial learning rate is set to $1e^{-4}$ with linear decay after 50 epochs. The USC-PFN consists of NGD and SIG, both have the same structure as Res-UNet [30], and the discriminator is from pix2pixHD [16]. In the loss functions, the $\lambda_r = 20$ and $\lambda_p = 0.25$ in the $\mathcal{L}_{ngd}$. The $\lambda_{scyc} = 1$, $\lambda_{adv}^G = 0.1$, $\lambda_{sr} = 50$, $\lambda_{gr} = 1$, and $\lambda_{cp} = 10$ in the $\mathcal{L}_{sig}^t$.

**Baselines.** We perform comparative evaluations of our model's performance, which includes the deformer and generator, against publicly available state-of-the-art methods. To this end, we leverage ten popular methods, including VITON[8], CP-VTON[3], Cloth-flow[9], CP-VTON+[27], ACGPN[4], DCTON[2], PF-AFN[6], ZFlow[31], SDAFN[28], and RT-VTON[13], as baseline methods for quantitative evaluation. In particular, we select seven most cutting-edge methods [27, 4, 2, 6, 28, 13, 7] for qualitative evaluation.

**Evaluation Metrics.** Quantitative evaluations are carried out in both paired and unpaired settings to compare our approach with baseline methods. For the paired setting, we use Structure Similarity (SSIM) [32] to evaluate the visual quality and diversity of the generated images. In the unpaired setting, as there is no ground truth available, FID is directly employed to evaluate the distributional similarity between the generated and real images. Note that we do not use Inception Score (IS) [33]

Table 1: Quantitative results of different methods on VITON. 'Def.' represents different warping methods. 'Syn.' represents different synthesizing paradigms. 'Parser' indicates whether the parser is used in the model during inference. The best result are in **bold** and the second best result are in blue.

| Methods | Publication | Def. | Syn. | Parser | SSIM ↑ | FID ↓ |
|---------|-------------|------|------|--------|--------|-------|
| VITON [8] | CVPR 2018 | TPS | IP | Y | 0.74 | 55.71 |
| CP-VTON [3] | ECCV 2018 | TPS | IP | Y | 0.72 | 24.45 |
| Cloth-flow [9] | ICCV 2019 | AF | IP | Y | 0.84 | 14.43 |
| CP-VTON+ [27] | CVPRW 2020 | TPS | IP | Y | 0.75 | 21.04 |
| ACGPN [4] | CVPR 2020 | TPS | IP | Y | 0.84 | 16.64 |
| DCTON [2] | CVPR 2021 | TPS | CC | Y | 0.83 | 14.82 |
| PF-AFN [6] | CVPR 2021 | AF | KD | N | 0.89 | **10.09** |
| ZFlow [31] | ICCV 2021 | AF | IP | Y | 0.88 | 15.17 |
| RT-VTON [13] | CVPR 2022 | MLS | IP | Y | - | 11.66 |
| SDAFN [28] | ECCV 2022 | AF | IP | N | 0.88 | 12.05 |
| USC-PFN (Ours) | This Work | MRF | SC | N | **0.91** | 10.47 |

− : official code or data are not provided. IP: in-painting; CC: cycle consistency; KD: knowledge distillation; SC: self-cycle consistency.

in the evaluation [34, 35], as it is only effective in datasets similar to ImageNet [36]. Specifically, a lower score of FID indicates a higher quality of the result.

## 4.1 Experimental Evaluation

**Qualitative Results.** In Figure 4, we perform visual comparison with seven methods, including SOTA in-painting methods [27, 4, 13, 28], the SOTA cycle consistency method [2], and SOTA knowledge distillation methods [6, 7]. In particular, we specifically select four striped garments to visually showcase the deformation effects of each method. It indicates that all methods can achieve approximate garment alignment and try-on synthesis. However, in terms of garment alignment, the TPS-based methods

Table 2: Analysis of time cost and computational complexity between ACGPN[4], DCTON[2], PF-AFN[6], and ours.

| Methods | Time | #Params | FLOPs | FPS |
|---------|------|---------|-------|-----|
| ACGPN [4] | ∼40h | 139M | 206G | 10 |
| DCTON [2] | ∼44h | 153M | 194G | 19 |
| PF-AFN [6] | - | **99M** | 69G | 34 |
| Ours | ∼32h | 140M | **46G** | **39** |

(columns 3, 4, and 5) exhibit excessive local fabric deformation and misaligned spatial correspondence. The AF-based methods (columns 6, 7, and 8) demonstrate overly flexible and unconstrained deformation effects, which are particularly noticeable in the first row. The MRF-based method (column 9) shows semi-rigid deformation effects in highly non-rigid body poses, such as misalignment in the arm region. In terms of try-on synthesis, the in-painting methods (columns 3, 4, 7, and 9) exhibit mismatches between the given garments and the synthesized try-on results due to parsing errors, particularly noticeable in the arms region. The cycle consistency method (column 5), being modular in nature, shows artifacts along the module boundaries. The knowledge distillation methods (columns 6 and 8) occasionally suffer from similar issues as the in-painting methods, both in terms of clothing deformation and try-on synthesis, due to irresponsible teacher knowledge. In contrast, our proposed method (last column) can achieve the most realistic results benefits from two key factors. Firstly, it leverages MRF's perception of human spatial structure to faithfully simulate the process of cloth deformation after try-on, capturing realistic fabric draping effects. Secondly, it utilizes the self-cycle consistency paradigm to accurately reconstruct the appearance of the garment worn on the body, all without the need for additional parsers or complex designs.

**Quantitative Results.** We evaluate the quantitative results of different methods under the same configuration for a fair comparison. We use SSIM for measuring structural similarity in the paired setting and FID for measuring distributional similarity in the unpaired setting. Table 1 presents our quantitative results, including the SSIM and FID scores of baseline methods [8, 3, 9, 27, 4, 2, 6, 31, 28, 13] and the proposed USC-PFN, on the VITON dataset. In terms of SSIM, USC-PFN

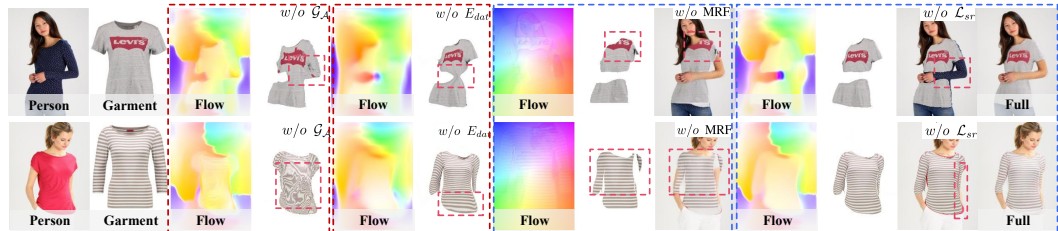

Figure 5: Ablation studies on the effect of NGD and SIG in the unpaired setting.

surpasses the SOTA in-painting method SDAFN [28] by 0.03, the SOTA knowledge distillation method PF-AFN[6] by 0.02, and the SOTA cycle consistency method DCTON [2] by 0.08. In terms of FID, it surpasses the SOTA in-painting method RT-VTON [13] by 1.19 and the SOTA cycle consistency method DCTON by 4.35. Although PF-AFN obtains a lower FID score, some of their results appear to be worse than ours [3]. Additionally, PF-AFN requires complex training with the assistance of the teacher model. The result demonstrates the superiority of our self-cycle consistency approach while showcasing the significant advantage of generating high-quality try-on images.

**Computational Complexity.** In addition, to demonstrate the superior performance of our network, we calculate the computational costs of ACGPN[4], DCTON[2], PF-AFN[6], and our method as shown in Table 2. On the same Tesla V100 GPU, we have an appropriate number of parameters and the fewest FLOPs (floating point operations), yet our inference speed is nearly four times faster than ACPGN and about twice as fast as DCTON. This demonstrates the real-time capability of our method and its ability to achieve high-quality visual performance with fewer computational resources. Note that our architecture can employ any encoder-decoder network, allowing for flexible selection of the network based on computational resources.

**Ablation Study.** In the ablation study, we evaluate the effectiveness of NGD and SIG in USC-PFN. Table 3 and Figure 5 provide both quantitative and qualitative results. Firstly, we demonstrate that the auxiliary deformer $\mathcal{G}_{\mathcal{A}}$ successfully disentangles the color, texture, and shape correlations in the pre-trained $\mathcal{G}_{\theta}$. The experiments further validate the necessity of the data term $E_{dat}$ in MRF, as it plays a crucial role in ensuring the accurate deformation of the garment. Moreover, we conduct an experiment by removing the entire MRF module to emphasize its essential in USC-PFN. Finally, the efficacy of $\mathcal{L}_{sr}$ in SIG is validated, emphasizing its critical role in supervising skin generation.

**Limitations and Discussion.** There are still limitations to USC-PFN that need to be addressed in future work. Firstly, due to the complex non-rigid deformation of garment images, it is challenging to achieve good convergence during end-to-end training; that is, integrating the garment deformer $\mathcal{G}_{\theta}$ into the image generator $\mathcal{G}_{\varphi}$ remains a difficult task. Secondly, accurately aligning garments with human poses is challenging due to the complexity and variability of human poses. Thus, finding a more optimal closed-form solution is necessary to solve the garment deformation task. Finally, we still require extensive supervision during the training, and it is still an urgent issue to find a solution to train a high-quality network with less supervision.

Table 3: Ablation studies of NGD and SIG in the unpaired setting. Lower score of FID indicates higher quality of results.

| NGD | FID ↓ | SIG | FID ↓ |
|---|---|---|---|
| $w/o$ $\mathcal{G}_{\mathcal{A}}$ | 42.23 | $w/o$ MRF | 14.28 |
| $w/o$ $E_{dat}$ | 30.53 | $w/o$ $\mathcal{L}_{sr}$ | 15.44 |
| Full NGD | **25.44** | Full SIG | **10.47** |

## 5 Conclusion

In this paper, we present USC-PFN, a parser-free virtual try-on network that utilizes a self-cycle consistency pipeline, which only uses one model to cyclically learn trying on different styles of

---

[3]See more visual comparisons in the supplementary material.

garments. To get a more natural and realistic garment alignment, USC-PFN first incorporates the Markov Random Field as a non-rigid deformation method, by enhancing the deformer's perception of human spatial structure, thereby mimicking the natural interaction between the garment and the person. In addition, unlike existing paradigms, our method does not require input information from parsers, teacher knowledge, or complex person representations for the generator. Instead, it solely relies on the garment and human images as input to train and infer try-on results. The result indicates that USC-PFN outperforms the state-of-the-art methods significantly in terms of subjective and objective evaluations, demonstrating its superior performance. In the future, we plan to extend this architecture to different tasks of image-to-image translation.

## Acknowledgments and Disclosure of Funding

This work was in part supported by the National Key Research and Development Program of China (Grant No. 2022ZD0160604) and NSFC (Grant No. 62176194), and the Key Research and Development Program of Hubei Province (Grant No. 2023BAB083), the Project of Sanya Yazhou Bay Science and Technology City (Grant No. SCKJ-JYRC-2022-76, SKJC-2022-PTDX-031), and the Project of Sanya Science and Education Innovation Park of Wuhan University of Technology (Grant No. 2021KF0031).

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
