# OpenReview forum: "Greatness in Simplicity: Unified Self-Cycle Consistency for Parser-Free Virtual Try-On"
_NeurIPS.cc/2023/Conference — NeurIPS 2023 poster_

### Official Review · Reviewer_99zV · 2023-07-05

**Soundness:** 2 fair
**Presentation:** 1 poor
**Contribution:** 2 fair
**Rating:** 5
**Confidence:** 4

**Summary:**

This work proposes a virtual try-on method. At inference time, no parser information (e.g., pose, segmentation map) is needed; given a person's image and a garment, the method is able to output a try-on image. To learn this model with a dataset of paired person and garment images, a deformation network is learned to warp the garment to match the person. Meanwhile, a generator is learned to combine the warped new garment and the person to create the try-on image. The generator is trained with a self-cycle-consistency objective, and the intuition is simple: a try-on image should be within the distribution of real images (adversarial loss), and trying on the person's original cloth on an already tried-on image should look like the original image (cycle-consistency loss). The method has multiple other components in it, making it hard to add this to the summary. The proposed method is compared with prior work and shows similar performance with the state of the art.

**Strengths:**

1. From the qualitative comparisons, using ground truth deformation field as supervision instead of other alternatives (appearance flow, TPS) seems to help reduce artifacts. Especially in the case where the clothes have stripes.

2. The work proposes a good idea to use an auxiliary deformer, that takes in only pose information (without RGB) of the person as input to predict the garment deformation. It makes sense that it will help generalizing perdiction when the garment is not paired with the person (the training set is all paired up). Since at inference time, only RGB input are given, this auxiliary deformer is used to generate pseudo ground truths for unpaired garment and person images. In short, auxiliary deformer "teaches" the final deformer to generalize better, which is a good idea.

**Weaknesses:**

1. In the related work section, it is mentioned that using parser information can introduce unreliable errors and artifacts. And even with teacher-student training, this issue can still lead to robustness issues. Meanwhile, I do not understand why the proposed method alleviates the issue. The auxiliary deformation network, which is essential to the method (see Fig. 5), is trained with parser information (dense pose). It will be great to clarify this.

2. Writing is hard to understand. It prevents readers from understanding what exactly the algorithm is. See the points below:

3. In line 189, the Lsec loss is refered to a citation without mentioning what exactly it is. Is it precisely the regularization term in Eq. 2?

4. The skin region is mentioned in the method section (s_p, p _s). How do you locate, define, or detect these regions? Same question holds for the content preservation loss: how do you locate the invariant content region (head, trouser, etc.)?

5. Is there any reason that the cycle-consistency loss does not need to be separated into skin, content and garment reconstruction loss? It seems compatible, and I don't understand why the loss function design cannot be applied symmetrically.

6. In figure 5, I do not understand the visualization in the left column of SIG w/o MRF. Having an explanation of the visualization is nice. Also, although I understand other visualziations are representing the deformation field, it is still good to mention it in the caption.

**Questions:**

1. The paper mentioned that prior work uses two CNNs to train models cyclically, while the proposed method uses one CNN to train a self-cycle. However, is there a reason why training one model directly with self-cycle consistency will not work? Also, I am guessing that the skin, content, and garment reconstruction loss is one of the important components to make training converge to a reasonable solution. It will be great to confirm this.

2. In my opinion, the draft needs a major update, especially in the method section. If the revision explains the design choices more clearer, I am happy to change my ratings.

**Limitations:**

1. The paper mentions that it is challenging to achieve good convergence during end-to-end training. Does this mean that a two-stage training (pretraining NGD then training SIG) is required for good results? Or does this mean in general the current method does not attain good convergence?

2. In line 290, what does it mean by closed-form solution here?

-----------------
Raised score to a borderline accept.

---

> ### Author Rebuttal · Authors · 2023-08-09
>
> > Thank you for your diligent efforts and valuable suggestions.
> ---
> ### Weaknesses:
> - **W-A1:** Thanks for your comments. Datasets consist of paired garment-person images $(g,p)$, where $p$ wearing $g$. Current methods use the person representation $I_p$ and $g$ to find dense spatial correspondences between $g,p$. However, $I_p$ lacks $p$'s depth and spatial info, leading to rigid deformation and fitting. We attempted to directly train main deformer $G_\theta$ using $(g,p)$, this is because extracted $p$'s features contain rich spatial and depth info, which flat $I_p$ does not possess. However, $p$ also introduce the same shape, color, and texture correlation features as those of $g$ (strong correlation), which hinders the generalizability of $G_\theta$.
> - Therefore, we train $G_A$ and $G_\theta$ using $(g,p)$, allowing the rich spatial and depth knowledge in $G_\theta$ to guide the convergence of $G_A$. Note that the strong correlation in $G_\theta$ is not learned by $G_A$, as $G_A$'s input ($I_p$, dense pose) does not contain clothing's shape, color, and texture info. Thereby alleviating the impact of irresponsible prior knowledge on $G_A$ stemming from erroneous $I_p$. Then, we retrain $G_\theta$ using $G_A$ to eliminate ineffective strong correlations in the latent feature space of $G_\theta$. In this way, we achieve an efficient $G_\theta$.
> ---
> - **W-A2:** We apologize for our writing issues. We previously lacked detailed explanations. Now, we have conducted extensive experiments and provided additional explanations to improve its readability.
> ---
> - **W-A3:** We apologize for any confusion caused by us. Yes, on lines 188 and 189, we clarify that $R_{sm}$ is implemented by $L_{sec}$, which indeed corresponds to the regularization term in Eq. (2).
> ---
> - **W-A4:** Thanks for your comments. Datasets contain garment $g$ and its mask $p_M$, person $p$ and its paring map $p_p$. $p_p$ includes several layers, the neck and arms layers are merged to form the skin layer, denoted as $p^n_p+p^a_p=p^s_p$. And the skin and garment layers are merged to form the agnostic layer, denoted as $p^s_p+p^c_p=p^u_p$. Then, $p_M$ that deforms along with $g$ is used as the new target clothing layer $\hat p_M$, and subtracting $\hat p_M$ from $p_p^u$ yields the new target skin layer $\hat p_p^s$ corresponding to $g$, i.e., $\hat p_p^s=p_p^u-\hat p_M$. $s_p$ is fixed skin region before and after trying on, which can be obtained by intersecting $\hat p_p^s$ and $p_p^s$, denoted as $s_p=(p_p^s \cap \hat p_p^s)\times p$. $\hat p'_s=\hat p_p^s \times \hat p$, where $\hat p$ is results generated in first stage. In this manner, the invariant content region (head, trouser, etc.) are also located by $p^h_p$, $p^t_p$ in $p_p$.
> - Note that during training and inference stages, we do not require the parser to estimate new human parsing as input. This perfectly aligns with the requirements of parser-free virtual try-on.
> ---
> - **W-A5:** Thanks for your comments. As shown in Fig. 1 (https://github.com/anony-conf/results-USC-PFN), we initially define a cycle-consistency loss $L_{scyc}$, which is employed to supervise the reconstruction of GT $p$. This global loss is used to preserve the structural and distributional consistency between the reconstructed $p'$ and $p$, ensuring a harmonious coordination of generated skin, garment, and the overall appearance. However, it cannot specially focus on the generation of localized body regions, particularly in first stage. Since there is no GT available to calculate $L_{scyc}$, we introduce only the skin, content, and garment reconstruction losses in first stage to optimize these regions that can be supervised. Furthermore, due to the weight sharing between the two stages, designing symmetrical loss functions becomes unnecessary.
> ---
> - **W-A6:** Yes, the left column of SIG w/o MRF is the deformation field $f$, indicating the effects after removing $f$ generated by NGD during the try-on synthesis stage of SIG.
> ---
> ### Questions:
> - **Q-A1:** Thanks for your comments. As shown in Fig. 1 (https://github.com/anony-conf/results-USC-PFN), in first stage, the try-on result $p'$ do not have corresponding GT in dataset. If we do not supervise this stage and directly use $p'$ as input for the second stage, it can be considered as a direct mapping from input $p$ to output $p$. However, our task requires achieving the cyclic mapping $p$→$p'$ and $p'$→$p$. Therefore, directly applying self-cyclic consistency training in the second stage will not work.
> - These reconstruction losses are crucial components, 1) there is a need to enforce a mapping of the outputs from the first stage into the domain of $p'$, requiring the formulation of local losses for the first stage; 2) the second reason is the same as **W-A5**.
> ---
> - **Q-A2:** We sincerely appreciate your professional and responsible comments and suggestions. Based on the feedback from all the reviewers, we have made major revisions to our manuscript, including the addition of the required experiments. Therefore, we earnestly implore that you reconsider the ratings of our paper.
> ---
> ## Limitations:
> - **L-A1:** Thanks for your comments. To demonstrate the necessity of two-stage training, we conducted end-to-end training experiments to demonstrate that this manner cannot achieve the same favorable convergence as the two-stage approach, both NGD and SIG did not converge, resulting in an extremely low SSIM score of 0.37 (see Fig. 8.5 (a)). We also computed the FID and KID values on the test set during training to reflect the convergence of our manner (see Fig. 8.5 (b)). This successfully validates all the points in **Q-A1**.
> ---
> - **L-A2:** The term "closed-form solution" in line 290 refers to our proposed self-cycle consistency pipeline.
> ---
> > We sincerely thanks for your dedicated efforts. We hope that our responses have provided you with new insights into our work. We look forward to you giving us a chance by increasing your approval of our paper.

---

> > ### Comment · Reviewer_99zV · 2023-08-14
> > **More clarifications**
> >
> > Thanks so much for the clarifications. Having these in the revised draft will strengthen the paper a lot. To confirm, are all the color-coded visualization in Fig. 5 deformation fields? Is the deformation prediction for SIG w/o MRF simply not performing well? Thanks!

---

> > > ### Author Response · Authors · 2023-08-15
> > >
> > > > Thank you very much for your professional and careful feedback. Yes, all the color-coded visualizations in Figure 5 represent deformation fields. In other words, it is the visual results of the deformation fields.
> > >
> > > > Regarding the SIG w/o MRF results, the visual results of SIG w/o MRF in Figure 5 are the worst, as the given clothing undergoes only slight deformation, resulting in very smooth stripes and severe misalignment of logos on the clothes.
> > >
> > > > Thank you once again for investing significant effort in reviewing our work. Your insightful and professional feedback has greatly improved the quality of our paper.

---

> > > > ### Comment · Reviewer_99zV · 2023-08-17
> > > >
> > > > Also, following up on the first weakness point:
> > > >
> > > > So if I understand correctly, prior work predicts a spatial deformation with an image representation (which is likely a feature vector), and the contribution of this work is to use a spatial feature instead. Moreover, this spatial feature is pretrained with dense pose information, which will be less correlated with textures and colors. Do you mind confirming this? Thanks!

---

> > > > > ### Author Response · Authors · 2023-08-18
> > > > > **Response to the first weakness point**
> > > > >
> > > > > > Thank you for your professional and thoughtful comments. During clothing deforming, there are two main differences between our work and previous works: **1)** In previous works, the input consists of a clothing image ($C$) and a person representation item ($I_p$) of person image $I$, such as human parsing maps (semantic segmentation maps) or pose heatmaps, to predict spatial deformations. As one can imagine, the person representation only contains rough information about human body shape distribution; for instance, flattened semantic information in human parsing maps, and rough shape information in pose maps. ***Thus, when $I_p$ is used as inputs, previous works lacked crucial 3D spatial information unique to person images, such as perspective changes, texture variations, shadows and lighting, depth of field, and more.*** Consequently, the warped results obtained mainly focus on deforming ($C$) to align with the clothing semantic layer of $I_p$, without considering whether the warped result corresponds naturally and reasonably to the spatial structure of the human body ($I$). This can be observed in our qualitative results, where the results of previous works show some cases where clothing fabric is stretched from the abdomen to the arm area.
> > > > >
> > > > > > **In contrast, our approach no longer relies on person representations as inputs. Instead, we use human images ($I$) that contain rich 3D spatial information as inputs.**
> > > > >
> > > > > - But why don't previous works take $I$ as inputs?
> > > > >
> > > > > > This is because the dataset only contains paired $(C, I)$, where $I$ is wearing clothing $C$. Hence, when taking $(C, I)$ as input, there is a strong correlation between $C$ and $I$, especially in terms of color, shape, and partial texture. Consequently, when training the deformer using $(C, I)$ as inputs, they become highly coupled. In other words, they deform well only when encountering paired images, while they fail when presented with randomly input clothing and human images.
> > > > >
> > > > > > Therefore, when $(C, I)$ is used as input, the significant challenge lies in eliminating the strong correlations from the extracted feature space, while retaining valuable features such as perspective changes, shadows, lighting, depth of field, and more. This allows the deformer to perceive the 3D spatial information of the human body.
> > > > >
> > > > > > Thus, the second difference arises: **2) Our work successfully eliminates strong correlations and optimizes the generated displacement field $f$ directly to optimize the model by employing the idea of Markov Random Fields (MRF).** To the best of our knowledge, previous methods have not yet resolved this problem.
> > > > >
> > > > > - How does our work eliminate strong correlations?
> > > > >
> > > > > > We initially co-trained $\mathcal G_\theta$ and an auxiliary deformer $\mathcal G_A$. Here, the input for $\mathcal G_\theta$ is paired $(C, I)$, whereas $\mathcal G_A$ takes in $C$ and a prior densepose descriptor $I_d$ as inputs.
> > > > >
> > > > > > The $I_d$ maps each pixel on the human body surface to corresponding body parts and pose information, encompassing labels for body parts as well as more detailed pose details. Therefore, it has no strong correlations with $C$, but it contains more human spatial information than semantic maps or pose point maps.
> > > > >
> > > > > > During the synchronized training of $\mathcal G_\theta$ and $\mathcal G_A$, due to the input $I_d$ of $\mathcal G_A$ still lacks depth and spatial information, we initially use the outputs of $\mathcal G_\theta$ to supervise $\mathcal G_A$, denoted as: $\mathcal G_\theta(C,I)=f$, $\mathcal G_A(C,I_d)=f^*$, calculating $loss[f, f^*]$ for $\mathcal G_A$, to supplement the lack of depth and spatial information in $\mathcal G_A$.
> > > > >
> > > > > > After training $\mathcal G_A$, we retrain $\mathcal G_\theta$ and allow $\mathcal G_A$ to supervise $\mathcal G_\theta$. This step aims to have $\mathcal G_A$ penalize the strong correlations in the latent feature space of $\mathcal G_\theta$ in order to eliminate its influence, i.e., calculating $loss[f, f^*]$ for $\mathcal G_\theta$. Note that both the supervision and penalization are achieved by directly computing the differences in $f$, a feature not present in other methods and aligned with the concept of MRF.
> > > > >
> > > > > > In summary, previous works utilized $C$ and person representations $I_p$ (semantic maps, pose keypoints) to predict deformation space, and optimizing deformer by minimizing the differences between $C$ and $I_p$. This severely compromised the network's ability to perceive human depth and spatial information. In contrast, our work directly uses paired $(C,I)$ as input (instead of use a spatial feature), effectively mitigating the highly coupled strong correlations (textures, shapes, colors, etc.) through $\mathcal G_A$. Furthermore, following MRF concept, we optimize the deformation space by directly supervising the deformation field.
> > > > >
> > > > > > I apologize once again for any confusion caused by us. If you have any further concerns, please reach out to us. Thank you!

---

> > > > > > ### Comment · Reviewer_99zV · 2023-08-21
> > > > > >
> > > > > > Thanks for the clarification! I will raise my score to borderline accept.

---

### Official Review · Reviewer_RUsU · 2023-07-10

**Soundness:** 3 good
**Presentation:** 2 fair
**Contribution:** 3 good
**Rating:** 4
**Confidence:** 4

**Summary:**

This paper proposes a new parser-free virtual try-on network (USC-PFN) to use only unpaired images as input to generate realistic try-on results.
To address the core warping problem in virtual try-on, it models the deformation field estiamtion by using the Markov Random Field. To train the try-on generator by using unpaired data, it proposes a self-cycle consistency pipeline.
Extensive  comparisons with the state-of-the-art methods on VITON benchmark demonstrate its superiority and the ablation study also shows the effectiveness of different modules in the proposed method.

**Strengths:**

- This paper explores the parser-free virtual try-on, which is challenging and with great significance for the image-based virtual try-on.

- For the first time, this paper introduces the Random Markov Field into the non-rigid garment deformer, which is quite different from the deformation module in previous methods.

- It proposed a novel self-cycle consistency pipeline for the training of the try-on generator by using unpaired images.

- The authors conduct extensive comparisons with the existing state-of-the-art methods to illustrate the superiority of the proposed methods. The ablation study also show the effectiveness of the different modules in the proposed method.

**Weaknesses:**

- One core technical contribution in this paper is that it model the garment warping by using the Markov Random Field. However, the authors do not provide an clear explanation about it. What is the difference between it and the widely used appearance flow? why can it outperform the TPS-based or appearance flow-based methods?

- The authos only conduct experiments on the VITON benchmark, in which the image resolution is quite low (256 x 192). However, most of the advanced methods focus on the higher resolution virtual try-on (e.g., 512 x 384, 1024 x 768), which are more closed to the real world
try-on scanerio.

- The writing is not straightforward amd several descriptions are a bit obscure. For example, (1) Although  the garment deformer and the auxiliary deformer receive different inputs, in Figure 3, they seems to take the same inputs (i.e., person image and the garment image). (2) In line131, it is confused how to obtain the deformation field $\tilde{f}$.

- Some decipt in the main paper is a bit overclaimed. In line 300, the authors claim that USC-PFN can solely rely on the garment and human image for training. However, it still required some human condition like human parsing, densepoe during training NGD and SIG.

**Questions:**

- For the non-rigid deformation module, is the pre-trained deformer necessary for the training of the final deformer? In my opinon, it is unnecessary to use supervision provided by the pre-trained deformer  (which is trained by using paired images) for the training of auxiliary deformer, since the ground truth deformation field is not necessary for the training of deformer (velidated in the previous works like PF-AFN[1], FS-VTON[2]). Once obtaining the auxiliary deformer, we could train the final deformer by using the unpaired images.

- When training the Self-cyclic Image Generator (SIG), several local region supervisions (i.e., skin loss, garment loss, preserved content loss) are employed to facilitate the self-supervised cyclic training, which should resort to the human parsing to obtain the specific local region. Thus, the parsing error might still affact the training procedure. However, some knowledge distillation method like PF-AFN[1] does not face with parsing error issue since they only use the global supervision when the the parser-free student network. My question is how does USC-PFN alleviate the influence derived the parsing error during training.

- Since the SIG can be trained with unpaired images, is it possible for USC-PFN to leverage large amout of unpaired images from Internet during the trianing of SIG? Will such the increasing in training data facilitate the performance of USC-PFN?

[1] Ge et al. "Parser-Free Virtual Try-on via Distilling Appearance Flows", CVPR 2021.

[2] He et al. "Style-Based Global Appearance Flow for Virtual Try-On", CVPR 2022.

**Limitations:**

Some limitations of this paper have been disscussed in the main paper. Other limitation I concern is the image resolution, since it is quite improtant for the real world try-on scanerio.

---

> ### Author Rebuttal · Authors · 2023-08-09
>
> > Thank you for your diligent efforts and valuable suggestions.
> ---
> ### Weaknesses:
> - **A1:** Thanks for your comments. Datasets consist of paired garment-person images $(g,p)$, where $p$ wearing $g$. Existing methods use the person representation $I_p$ and $g$ to find dense spatial correspondences between $g$ and $p$. They adjust garment shape to minimize shape differences between deformed garment $\hat g$ and the Ground Truth (GT, garment on $p$). Due to the lack of GT flow field, appearance flow (AF) -based methods indirectly predict the flow field $F$ by the supervision of warped garment's shape ($\hat g$) [19], i.e., calculate the loss between $\hat g$ and GT. Two issues arise: 1) $I_p$ lacks $p$'s depth and spatial info, leading to rigid garment deformation and unrealistic fitting. 2) Optimizing $F$ indirectly by optimizing $\hat g$'s shape with pixel similarity disregards structural and depth correlations between $\hat g$ and $p$, causing excessive deformation. Similarly, TPS-based methods predict control points to calculate the same flow field. So, the drawbacks of both methods are the same. And TPS is more rigid.
> - Our method tackled the problems. 1) We use $p$ instead of $I_p$ as input to input depth and spatial info. But $p$ and $g$ are paired, to solve this issue, 2) we introduce Markov Random Field (Eq. 1) for clothing deformation, which supervises the estimated deformation field $f$ by GT field $\bar f$. However, $\bar f$ does not exist in the dataset. Therefore, we introduce the auxiliary deformer $G_A$ to learn depth and spatial info from $p$. $G_A$, pretrained with densepose descriptor $p_d$, is used to remove irrelevant priors (color, texture, and shape features of $p$'s garment) in extracted features of $p$, ensuring only depth and spatial info remains. Thus, GT field $\bar f$ can be generated by $G_A$.
> - In summary, our approach directly eliminates relevant info from the feature of $p$ containing rich spatial and depth info, and employs MRF principles to directly supervise $f$, enhancing the model's spatial awareness. We provide both qualitative and quantitative results on low-resolution and high-resolution datasets in **A2**, to demonstrate the effectiveness of our deformer.
> ---
> - **A2:** Thanks for your suggestion. To demonstrate the effectiveness of our method, we have added qualitative and quantitative results of both NGD and SIG on VITON-HD dataset in https://github.com/anony-conf/results-USC-PFN , Sec. 1 to 5.
> ---
> - **A3:** (1) We sincerely apologize for the ambiguity in Fig. 3. There indeed are differences in their inputs, as explained in section 3.2 and **A1**. The garment deformer takes person $p$ and garment $g$ as inputs, while the auxiliary deformer takes the densepose descriptor $p_d$ of $p$, and $g$ as inputs. We have revised Fig. 3 to enhance its clarity. (2) The deformation field $f$ is directly generated by the $G_\theta$, i.e., $G_\theta(g,p)=f$, and $\hat g$ is obtained by bilinear sampling of $g$ using $f$.
> ---
> - **A4:** We deeply apologize for our inaccurate description. We intended to convey that USC-PFN can indeed be trained and inferred solely based on garment and person images **as input**.
> ---
> ### Questions:
> - **A1:** Thanks for your professional suggestion. Our auxiliary deformer $G_A$ is necessary, this is because we attempted to directly train the main deformer $G_\theta$ using paired ($g$, $p$), but features extracted from $p$ contain abundant shape, color, and texture correlations with $g$, which hinders the generalizability of $G_\theta$. Consequently, we had to separately train $G_A$ to eliminate latent shape, color, and texture correlations from $p$'s features, enabling feasible training with paired images. The specific process is detailed in above **A1**.
> ---
> - **A2:** Thanks for your insightful comments. Firstly, we employ a global self-cycle consistency loss $L_{scyc}$, which does not introduce parsing errors. For skin loss, we utilize a pre-trained SR trained using perceptual loss to generate skin regions, it produces clean skin outputs even when there are cloth-related impurities in the input. This is fully in accord with the consistency of the distribution of skin in perceptual loss. For garment loss, it is only used in first phase. We segment the garment area of $\hat g$ and $p$ using the mask of $\hat g$. As our network's input is also segmented $\hat g$, this loss ensures direct $\hat g$ output by SIG, so it only possibly introduces white boundaries caused by parsing errors. We address this via $L_{scyc}$ in second phase, which enforces supervision via GT without such boundaries. Preserved content loss is used to preserve the fixed region, it is penalized by skin loss and garment loss for regions that are incorrectly segmented. Even if preserved content is incomplete, SIG's extensive training with massive data mitigates the impact.
> - If possible, please refer to Sec. 8.4 (link above) for a more comprehensive explanation.
> ---
> - **A3:** Thanks for your constructive comments. During training of SIG, USC-PFN can leverage large amout of unpaired images from Internet. This facilitates the performance of USC-PFN. We augmented the VITON-HD dataset with additional VITON data, see the table below. The experimental results validate your point.
>
> |Methods|SSIM ↑|FID ↓|KID ↓|
> |:--:|:--:|:--:|:--:|
> |Ours (VITON-HD)|0.901|9.08|0.142|
> |Ours (VITON-HD + VITON)|0.906|9.01|0.131|
> ---
> ### Limitations:
> - **A1:** Thanks for your suggestion. We have added experiments on the high-resolution VTON-HD dataset , and specific qualitative and quantitative results can be found in https://github.com/anony-conf/results-USC-PFN , Sec. 1 to 5.
> ---
> > We sincerely thanks for your dedicated efforts. We hope that our responses have provided you with new insights into our work. We look forward to you giving us a chance by increasing your approval of our paper. If you have more questions or encounter broken links, please comment, and we'll assist you quickly.

---

> > ### Comment · Reviewer_RUsU · 2023-08-19
> > **Official Comment by Reviewer RUsU**
> >
> > Thanks for the detailed response. However, I still feel confused about some technical details.
> >
> > First, What is the essential difference between the proposed Markov Random Field and the widely used appearance flow? After reading the paper and the authors' rebuttal, I argue the Markov Random Field is quite similar to the appearance flow expect for its network inputs (i.e., receiving image rather than the person representation), training strategy (i.e., using unpaired images as training data, introducing an auxiliary deformer),  and the loss functions.
> >
> > Second, in the authors' rebuttal, the authors argue the person representation used in the previous appearance flow-based deformation methods lack depth and spatial information. However, methods like PF-AFN and FS-VTON takes densepose pose as the deformation netowork's input, which I argue can also provide the depth and spatial information.
> >
> > Third, the authors seem to misunderstand my concern in Question 1. I agree with the author that using an auxiliary deformer is necessary to provide pseudo gt. My question is why should we use the paird image to pre-train the deformer? (as mentioned in line 163-164). In my opinion, given the pseudo gt deformation field, the deformer could be directly trained by using the unpaired images.
> >
> > Forth, What is the specific implementation of $D$ in equation 1?

---

> > > ### Author Response · Authors · 2023-08-20
> > >
> > > > **A1:** Thanks for your comments. Markov Random Field (MRF) is widely employed in image registration, we extend this concept to the treatment of deformation fields ($f$), where each element (data term) of $f$ is associated with unary clique and bi-clique, representing the state of the element itself and its relationships with its neighboring elements, respectively. Ideally, accurately estimating the best state for each element ensures optimal alignment, which can solve occlusions. Effectively managing the relationships between elements and their neighborhoods can address excessive deformations.
> > >
> > > > For the unary clique, we optimizes each data term by minimizing the sum of absolute differences (SAD) between the estimated $f$ and ground truth (GT) $\bar f$. However, due to the absence of GT $\bar f$ in datasets, it is not feasible to directly compute SAD. Therefore, we employ an auxiliary $\mathcal G_A$ to provide us with pseudo-GT $\bar f$. $\mathcal G_A$ takes $C$ and the densepose $I_d$ (without UV map) of the person $I$ wearing $C$ as inputs. However, $I_d$ is the semantic map of $I$, which lacks ***depth information and 3D spatial information of $I$. Specifically, perspective changes, texture variations, shadows and lighting, and depth of field present in $I$ are absent in the $I_d$.***
> > >
> > > > **A2:** Similarly, although PF-AFN and FS-VTON also use densepose $I_d$ (without UV) as input, $I_d$ lacks aspects present in person images such as perspective changes, texture variations, shadows and lighting, and depth of field. Therefore, PF-AFN, FS-VTON, and our $\mathcal G_A$, all lack the 3D spatial and depth information of $I$. So, the warped results obtained by theirs teacher models mainly focus on deforming $C$ to align with the clothing and arm semantic layers of $I_d$, without considering whether the warped result corresponds naturally and reasonably to the spatial structure of the human body. This can be observed in our qualitative results, where the results of these methods show some cases where clothing fabric is excessively stretched from the abdomen to the arm area, which also leads to occlusions.
> > >
> > > > To address this, we initially co-trained our main deformer $\mathcal G_\theta$ and $\mathcal G_A$. Here, $\mathcal G_A$ takes $C$ and densepose $I_d$ as inputs, $\mathcal G_\theta$ takes paired $(C, I)$ as inputs to extract depth and 3D spatial information of $I$. We use the outputs of $\mathcal G_\theta$ to supervise $\mathcal G_A$, denoted as: $\mathcal G_\theta(C,I)=f$, $\mathcal G_A(C,I_d)=\bar f$, calculating $loss[f, \bar f]$ for $\mathcal G_A$, **to supplement the lack of depth and 3D spatial information in $\mathcal G_A$.**
> > >
> > > > **A3:** However, pseudo-GT $\bar f$ is never as accurate as the real GT, which can limit the improvement of clothing deformation, and the usage of flawed $I_d$ as input can impact the deformation outcomes. As input, $I$ does not have these issues. However, there is a strong correlation between paired $C$ and $I$ because their clothing shares the same colors, textures, and shapes. This means that a deformer trained using $(C, I)$ exhibits high coupling. In other words, it becomes ineffective when faced with unpaired data. Additionally, utilizing unpaired data as input can address the above issue. However, unpaired data lacks local awareness between clothing and the human body. In other words, the deformer doesn't know which part of the clothing corresponds to the neckline and which part corresponds to the sleeves. Paired data, on the other hand, can ensure this alignment based on color and texture consistency. As a result, our approach achieves good deformation results even on complex curved arms, and the alignment of necklines is also relatively accurate.
> > >
> > > > To combine the advantages, we incorporate both paired and unpaired data in the architecture of the entire cycle training. Specifically, the training of deformer is divided into two stages: the first stage employs paired data to retain the ability of $\mathcal G_\theta$ to perceive the depth and 3D spatial information, while the second stage utilizes unpaired data as input (this stage is similar to PF-AFN and FS-VTON, with the difference that we use cycle consistency to continuously adjust pseudo-GT $f$, thereby refining deformations) to overcome the limitations imposed by the pseudo-GT $\bar f$.
> > >
> > > > In the first stage, after training $\mathcal G_A$, we retrain $\mathcal G_\theta$ and allow $\mathcal G_A$ to supervise $\mathcal G_\theta$. This step aims to have $\mathcal G_A$ penalize the strong correlations in the latent feature space of $\mathcal G_\theta$, i.e., calculating $loss[f, \bar f]$ for $\mathcal G_\theta$.
> > >
> > > > For the bi-clique, we apply improved regularization to constrain $f$.
> > >
> > > > **A4:** The $D$ is implemented by the L1 norm and perceptual loss for the second stage stated in A3 above.
> > >
> > > > I apologize once again for any confusion caused by us. If you have any further concerns, please reach out to us. Thank you!

---

> > > > ### Comment · Reviewer_RUsU · 2023-08-20
> > > > **Official Comment by Reviewer RUsU**
> > > >
> > > > Thanks for your comprehensive response. My concerns have been addressed. I will raise my rating after the discussion period.
> > > >
> > > > By the way, I suggest the authors to add the above analysis in the revision (in main paper or appendix) to make the paper more straightforward to its reader.

---

### Official Review · Reviewer_XB1J · 2023-07-14

**Soundness:** 2 fair
**Presentation:** 2 fair
**Contribution:** 1 poor
**Rating:** 5
**Confidence:** 5

**Summary:**

The paper addresses the challenges in generating high-quality virtual try-on images, specifically focusing on non-rigid garment deformation and unpaired garment-person images. Existing methods rely on disentangling garment domains with the aid of "teacher knowledge" or dual generators, which can limit the quality of try-on results. Additionally, current garment deformation techniques struggle to mimic natural interaction between garments and the human body, resulting in unrealistic alignment effects. To overcome these limitations, the authors propose a Unified Self-Cycle Consistency for Parser-Free virtual try-on Network (USC-PFN). USC-PFN utilizes a single generator and incorporates Markov Random Field for more realistic garment deformation.

**Strengths:**

- The paper is easy to understand.
- The experiment results seems okay, but not excellent compared to the baselines.
- The proposed method requires less computational costs compared to the baselines such as ACGPN and DCTON during inference.

**Weaknesses:**

- The primary limitation of this paper lies in the lack of significant performance improvement compared to the existing virtual try-on baselines. Particularly, when examining the performance comparison in Table 1, it raises doubts about the meaningful enhancement in terms of FID, which is even higher than that of PF-AFN. Furthermore, the disparity in frames per second (FPS) between PF-AFN and the proposed approach is not significant.
- The proposed self-cyclic image generator is inspired by StarGAN; however, it is unclear what specific advantages it offers. It appears to train a generator with clothing as a condition instead of employing other models that utilize cycle consistency loss. Nevertheless, it is uncertain whether this approach is novel and impactful enough to be presented at a top-tier conference..
- The newly proposed non-rigid garment deformer based on Markov Random Field lacks sufficient examples to demonstrate its superior warping capabilities compared to existing warping methods. Various virtual try-on methods have also proposed different approaches to enhance the performance of warping. However, it remains unclear how the proposed method establishes its superiority in this regard qualitatively and quantitatively.
- Most recent virtual try-on models have demonstrated their performance on high-resolution datasets such as VITON-HD [1] or Dresscode [2]. Additionally, parser-free virtual try-on models like DC-VTON have also shown their performance on images of around 512x384 resolution. However, this paper is lacking references to related works and only presents evidence of its performance on the low-resolution VITON dataset, which is somewhat disappointing.

[1] VITON-HD: High-Resolution Virtual Try-On via Misalignment-Aware Normalization, CVPR 2021
[2] Dress Code: High-Resolution Multi-Category Virtual Try-On, ECCV 2022

**Questions:**

Does the paper include a comparative analysis between the proposed warping method and other existing warping methods? I contend that it is imperative to conduct a direct comparison, encompassing both qualitative and quantitative evaluations, with well-established warping techniques such as TPS transformation and Appearance Flow.

**Limitations:**

Mentioned above.

---

> ### Author Rebuttal · Authors · 2023-08-09
>
> > Thank you for your diligent efforts and valuable suggestions.
> ---
> ### Weaknesses:
> - **W-A1:** Thanks for your comment. Our method significantly outperforms both RT-VTON (CVPR 2022) and SDAFN (CVPR 2022), which are more advanced than PFAFN. Moreover, our method outperforms PFAFN in KID and SSIM. We compare the performance improvements of the three methods over PFAFN; see Table 1.
>
> |Table 1|SSIM↑|FID↓|KID↓|SSIM*|FID*|KID*|
> |:--:|:--:|:--:|:--:|:--:|:--:|:--:|
> |PFAFN|0.89|10.06|0.264|/|/|/|
> |RT-VTON|-|11.66|-|-|- 14.2%| - |
> |SDAFN|0.88|12.05|-|- 1.1%|- 18.0%| - |
> |**Ours**|**0.91**|10.47|**0.249**|+ 2.2%|- 4.1%| + 5.7%|
> - In Table 1, our method shows the most significant improvement. In addition, we introduced a publicly available augmented VITON test set (see Table 2) and the high-definition VITON-HD dataset for a comprehensive and fair evaluation.
>
> |Table 2|SSIM ↑|FID ↓|
> |:--:|:--:|:--:|
> |ACGPN|0.81|20.75|
> |Cloth-flow|0.86|13.05|
> |PF-AFN|0.87|12.19|
> |**Ours**|**0.90**|**10.48**|
> - In Table 2, the SSIM of our method reaches 0.90, an increase of 3.4% compared to PFAFN. The FID is 10.48, which is a reduction of 14% compared to PFAFN. Next, we counted the performance of baseline methods on VITON-HD (512$\times$ 384) (see Table 3).
>
> |Table 3|Pub.|Parsing as input|SSIM ↑|FID ↓|KID ↓|
> |:--:|:--:|:--:|:--:|:--:|:--:|
> |CP-VTON|ECCV 2018|Y|0.791|30.25|4.012|
> |ACGPN|CVPR 2020|Y|0.858|14.43|0.587|
> |DCTON|CVPR 2021|Y|0.810|15.55|/|
> |VITON-HD|CVPR 2021|Y|0.843|11.64|0.300|
> |HR-VITON|ECCV 2022|Y|0.878|9.90|0.188|
> |**Ours**|/|**N**|**0.899**|**9.10**|**0.159**|
> - In Table 3, our approach has also surpassed the SOTA method, HR-VITON, this demonstrates the significant improvements in performance, highlighting the significance of our method. **The qualitative results on the VITON-HD are in Fig. 4 (https://github.com/anony-conf/results-USC-PFN).**
> - Furthermore, regarding FPS, we mentioned in line 275 of the paper that the parameters, FLOPs, and FPS of our network can vary with the used network. We intentionally reduced the *ngf* parameter in NGD (see Table 4), both parameters and FLOPs are significantly reduced.
>
> |Table 4|#Params|FLOPs|FPS|FID-Clothing ↓|SSIM-TryOn ↑|
> |:--:|:--:|:--:|:--:|:--:|:--:|
> |ACGPN|139M|206G|10|42.10|0.84|
> |DCTON|153M|194G|19|42.80|0.83|
> |PF-AFN|99M|69G|34|22.81|0.89|
> |Ours (All ngf=64)|140M|46G|39|18.70|0.91|
> |Ours (NGD ngf=32)|**87M**|**29G**|**40**|**18.85**|**0.91**|
> - We also calculated the complexity on VITON-HD (see Table 5).
> |Table 5|Weight|#Params|FLOPs|FPS|
> |:--:|:--:|:--:|:--:|:--:|
> |VITON-HD|588M|154M|1689.7G|3.76|
> |HR-VITON|586M|148M|1555.4G|4.09|
> |**Ours**|**334M**|**87M**|**467.9G**|**22.19**|
> ---
> - **W-A2:** Our method offers a novel viewpoint for virtual try-on tasks. In comparison to previously proposed solutions like disentangled cycle consistency [A] and knowledge distillation [B], its advantages are summarized as:
> 1) [A] uses dual-network during training, but only uses one network for inference, leading to convergence challenges and increased computational overhead. In contrast, our method achieves efficient convergence with a single shared-weight network, requiring fewer parameters and FLOPs.
> 2) [A] relies on human parsing as input during both training and inference, making the results sensitive to incorrect parsing. In [B], teacher network training and inference depend on the parser, where erroneous human parsing in teacher knowledge can mislead the student network. Our approach only takes garment and person images as input, thus avoiding above issues.
> 3) [A] employs disentangled cycle consistency to segment clothing and skin, which may cause artifacts at boundaries after their synthesis. Our method directly uses person images as input, avoiding this issue.
> 4) [A] introduces a multi-encoder network for feature extraction, adding complexity. [B] employs a complex clothing deformator for clothing deformation. Our approach can utilize any generator as our training network, achieving results surpassing [A] and [B].
> 5) [A] uses a globally deformable STN network for clothing deformation. [B] employs less controllable appearance flows for clothing deformation. We introduce an MRF-based clothing deformator achieving SOTA performance.
> 6) Our method outperforms the methods based on [A] and [B] in terms of realism, algorithmic complexity, model size, FPS, and generalization.
> - In summary, our approach possesses significant advantages and impact to offer a novel solution for virtual try-on tasks.
> ---
> - **W-A3:** We have added qualitative and quantitative results on VITON (see Table 6) and VITON-HD (see Table 7) datasets. Qualitative results are shown in Fig. 3 and 4 (https://github.com/anony-conf/results-USC-PFN).
>
> |Table 6|Pub.|Warping|FID-P ↓|KID-P ↓|FID-UP ↓|KID-UP ↓|
> |:--:|:--:|:--:|:--:|:--:|:--:|:--:|
> CP-VTON|ECCV 2018|TPS|43.95|2.233|42.13|2.112|
> ACGPN|CVPR 2020|TPS|42.10|2.009|41.48|2.048|
> DCTON|CVPR 2021|TPS|42.80|2.170|42.19|2.126|
> PFAFN|CVPR 2021|AF|22.81|0.785|23.90|0.860|
> SGAFN|CVPR 2022|AF|20.07|0.552|20.38|0.481|
> **Ours**|/|MRF|**18.70**|**0.390**|**19.50**|**0.355**|
> ---
> |Table 7|Pub.|Warping|FID-P ↓|KID-P ↓|FID-UP ↓|KID-UP ↓|
> |:--:|:--:|:--:|:--:|:--:|:--:|:--:|
> |VITON-HD|CVPR 2021| TPS|32.968|1.407|32.93|1.353|
> |HR-VITON|ECCV 2022|AF|25.499|0.926|24.826|0.759|
> |**Ours**|/|MRF|**19.060**|**0.418**|**22.861**|**0.504**|
> ---
> - **W-A4:** Thanks for your suggestion. We have added qualitative and quantitative experiments on the high-resolution dataset VITON-HD with a resolution of $512\times 384$, as shown in Table 3, and Fig. 3 and 4 (https://github.com/anony-conf/results-USC-PFN).
> ---
> ### Questions:
> - **Q-A1:**  We have added direct comparison experiments with well-established warping techniques such as TPS and Appearance Flow, please refer to **W-A3**.
> ---
> > Thanks for your dedicated efforts. We hope that our responses have provided you with new insights into our work. We look forward to you giving us a chance by increasing your approval of our paper.

---

> > ### Comment · Reviewer_XB1J · 2023-08-14
> > **Question on Table3 and Table7**
> >
> > I am grateful to the authors addressing my concerns effectively through various experiments in the rebuttal phase. If these experiments are adequately incorporated into the revised version of the paper, it is anticipated that they will enhance the quality of the paper significantly.
> >
> > I have some questions about the different outcomes observed between the Table 3 and Table 7. In Table 3, FID and KID for HR-VTON appear to align with the values reported in the original paper of HR-VTON. However, the results of Table 7 (in both paired and unpaired settings) are different from those of Table 3. The results presented in Table 3 appear to have been measured under the same conditions as HR-VTON. It would be beneficial to provide a detailed explanation of the method employed to measure the results presented in Table 7. Additionally, there is a notable disparity between the FID and KID metrics in Table 3 compared to the corresponding metrics in Table 7. It is recommended to offer a clear elucidation regarding the factors that have contributed to such divergent outcomes.
> >
> > Moreover, the proposed model demonstrates substantial improvements over VITON-HD and HR-VTON models in terms of FPS and FLOPS. It would be insightful to understand which specific module within the proposed architecture has significantly reduced the computational cost, whether it is attributed to the proposed warping module or the potential computational efficiency achieved within the image generator. Conducting an analysis to elucidate this aspect would further enhance the comprehensibility of the advancements achieved by the proposed model.

---

> > > ### Author Response · Authors · 2023-08-15
> > > **Answer on Table 3 and Table 7, and Computational Complexity**
> > >
> > > > **A1:** Thank you very much for your professional and careful feedback. I sincerely apologize for any confusion caused by our unclear presentation. We would like to clarify that **Table 3 is the quantitative results table for the entire network** on VITON-HD dataset, i.e., the quantitative results of the final virtual try-on result images. Apart from our data, the remaining data in the table were obtained from the official HR-VITON paper. **Table 7, similar to Table 6, is the quantitative results table for the non-rigid garment deformer** on VITON-HD dataset, i.e., the quantitative results of the warped garment images. Therefore, ***there is a notable disparity between the FID and KID metrics in Table 3 compared to the corresponding metrics in Table 7***, as Table 3 and Table 7 correspond to the Self-cyclic Image Generator (SIG) and Non-rigid Garment Deformer (NGD), respectively.
> > >
> > > > Furthermore, since the official papers of VITON-HD and HR-VITON did not include quantitative results for the garment deformer, we obtained their official codes and weights, and evaluated the garment deformation results under the same configurations as VITON-HD and HR-VITON. We apologize again for any confusion caused by our unclear table explanations.
> > >
> > > > **A2:** Thank you very much for your professional and constructive suggestions. Our work primarily proposes a new solution distinct from inpainting, cycle consistency, and knowledge distillation, which we term as "self-cycle consistency." Please refer to Figure 1 in the paper for more details.
> > >
> > > > Our proposed framework does not rely on the parser during the inference stage, currently, only the Knowledge Distillation architecture can achieve this. In knowledge distillation architecture, the parser-based teacher model imparts prior knowledge by providing the generated try-on results as pseudo ground truth or pseudo unpaired input to the student model.
> > >
> > > > Differently, our framework employs weight sharing in the two cyclic stages, allowing the try-on results generated in the first stage to be used as input for the second stage to reconstruct the person image achieving consistency. This is distinct from traditional cycle consistency, as implemented in DCTON [2], which has the following features: 1) It employs two non-shared dual networks for training, while only using one during inference (bearing some resemblance to knowledge distillation architecture). Simultaneously training two dual networks is challenging due to one network lacking ground truth. 2) It relies on human parsing as input, necessitating an additional parser during inference to generate corresponding human parsing for input. 3) Its architecture includes a parser (mask prediction network [2]) to generate clothing and skin masks, but flawed human parsing has been proven to lead to erroneous try-on results [6]. These three aspects can be considered shortcomings of [2], which our framework does not have.
> > >
> > > ---
> > >
> > > > Our framework consists of the Self-cyclic Image Generator (SIG) and the Non-rigid Garment Deformer (NGD). Due to the nature and features of our architecture, theoretically, any encoder-decoder network (such as ResUnet, Unet) can serve as our SIG and NGD. In our experiments, we used ResUnet for them, without incorporating any complex modules. Therefore, ResUnet can be replaced with a lighter Unet, significantly reducing computational costs without compromising performance.
> > >
> > > > On the other hand, HR-VITON incorporates a complex Try-On Condition Generator with carefully designed Feature Fusion Blocks and Condition Aligning to deform clothing, addressing occlusions and unnatural deformations. However, our experimental results demonstrate that our approach effectively resolves these issues using a standard ResUnet, and the results obtained are superior to HR-VITON. Similarly, its Try-On Image Generator employs a complex structure, resulting in higher parameters, FLOPs, and lower FPS. In contrast, our approach employs ResUnet or Unet for clothing deformation and try-on image generation, achieving better results than HR-VITON while significantly improving computational efficiency.
> > >
> > > - Therefore, we would like to clarify that our contribution lies in proposing an efficient architecture rather than meticulously designing specific modules to enhance result quality. Extensive experimentation substantiates the effectiveness of the architecture we have proposed.
> > >
> > > **We apologize for our extensive elaboration above, as we genuinely intended to address your concerns. Once again, we sincerely thank you for your careful review and professional feedback.**

---

> > > > ### Comment · Reviewer_XB1J · 2023-08-21
> > > >
> > > > Thanks for the authors' response. Since my concerns have bee addressed, I raise my score.
> > > >
> > > > However, it is necessary to update the main manuscripts a lot, to add the contents addressed in the rebuttal, including other reviewers' comments.

---

### Official Review · Reviewer_ugBm · 2023-07-25

**Soundness:** 3 good
**Presentation:** 3 good
**Contribution:** 3 good
**Rating:** 6
**Confidence:** 2

**Summary:**

The paper presented a system for image based virtual try-on. The main contribution consists (1) a parser free virtual try-on network trained with unpaired data; (2) a MRF based deformation estimation network; (3) a cycle consistency based training method. The paper performed experiments on the VITON Zalando dataset. The paper is evaluated with several prior work as baselines, including VITON[8], CP-VTON[3], Cloth-flow[9] etc. The main evaluation metrics are SSIM for pair data and FID for data with no ground truth. In terms of quantitative performance, the proposed method achieves better or comparable results than previous art. In terms of qualitative results, the generated images are visually comparable, and some times slightly better than previous work.

**Strengths:**

+ The paper is addressing an interesting problem.

+ The main idea is interesting and seems novel to me. Although cycle consistency are not new ideas and it has been used in previous work on virtual try on, the overall combination of ideas seems still new in this specific domain.

+ Extensive experiment shows state-of-the-art quantitative results. The qualitative results have less deformation artifact than previous work.

**Weaknesses:**

I'm not an expert on virtual try-on so I will mainly rely on the other reviewers opinion for a more informative feedback about the weakness of the paper. On feedback I have is that the results are often overly smooth on the garment area. The paper showed a lot of results of clothing with stripe pattern and the stripes are often very smooth, and not reflecting the pose and shape of human body very well.This is perhaps due to the way that the deformation is learned. It would be great to discuss the potential cause of this over-smoothness and also perform ablations studies.

**Questions:**

Please address the aforementioned questions.

**Limitations:**

The limitation is discussed in section 4 and seems sufficient to me.

---

> ### Author Rebuttal · Authors · 2023-08-09
>
> > Thank you for your diligent efforts and valuable suggestions in the peer review process.
> ---
> ### Strengths:
> **S-A:** Thanks for your comments. There is a significant difference between our main idea and the disentangled cycle consistency approach [A]. The differences are summarized as follows:
>
> 1) [A] uses dual-network during training, but only uses one network for inference, leading to convergence challenges and increased computational overhead. In contrast, our method achieves efficient convergence with a single shared-weight network, requiring fewer parameters and FLOPs.
>
> 2) [A] relies on human parsing as input during both training and inference, making the results sensitive to incorrect parsing. Our approach only takes garment and person images as input, thus avoiding above issue.
>
> 3) [A] employs disentangled cycle consistency to segment clothing and skin, which may cause artifacts at boundaries after their synthesis. Our method directly uses the whole person image as input, thus avoiding this issue.
>
> 4) [A] introduces a multi-encoder network for feature extraction, adding complexity. Our approach can utilize any generator as our training network, achieving results surpassing [A].
>
> 5) [A] uses a globally deformable STN network for clothing deformation. We introduce an MRF-based clothing deformator achieving SOTA performance.
>
> 6) Our method outperforms [A] in terms of realism, algorithmic complexity, model size, FPS, and generalization.
> - In summary, our approach possesses significant advantages and impact to offer a novel solution for virtual try-on tasks.
>
> ### Weaknesses:
> - **W-A1:** Thanks for your comments. In fact, this task is based on deep learning for image generation, thus sharing commonality with other image generation tasks. In comparison to previous virtual try-on literature, our paper provides comprehensive qualitative and quantitative experiments, comparing against state-of-the-art methods [2,3,4,6,7,8,9,13,27,28,31], to demonstrate the effectiveness of the proposed framework. The novelty of our approach is outlined in the introduction and related work sections. Moreover, we have adequately addressed and supplemented relevant experiments raised by other reviewers. Therefore, we sincerely appreciate your impartial and responsible evaluation. We have conducted experiments on selected state-of-the-art literature to showcase the comprehensiveness of our experiments, and the results are as follows.
>
> |Methods|Publication|Number of Baselines|Qualitative / Quantitative Results on Clothing Deformation|High-Resolution Dataset VTON-HD|
> |:--:|:--:|:--|:--:|:--:|
> |[2]|CVPR 2021|5 ( [ 3, 4, 8, 15, 27] )         |×/×|√|
> |[6]|CVPR 2021|5 ( [ 3, 4, 9, 11, 27] )         |×/×|√|
> |[28]|ECCV 2022|6 ( [ 3, 4, 6, 9, 11, 31] )     |×/×|×|
> |[7]|CVPR 2022|8  ( [2, 3, 4, 6, 8, 9, 27, 31] )|×/×|×|
> |[13]|CVPR 2022|3  ( [27, 4, 2] )               |√/×|×|
> |Ours|This Work|10                              |Added/Added|Added|
> - As can be observed, the experiments conducted for our method are thorough and comprehensive.
> ---
> - **W-A2:** Sorry for your confusion. Firstly, the selection of a substantial number of striped garments in the experiments was done to highlight the detailed and efficient performance of the newly proposed clothing deformation algorithm, MRF, in clothing deformation. This is because choosing clothing with large solid color blocks and no patterns would not showcase the significant pixel displacement caused by excessive deformation, whereas striped patterns allow for the clear visualization of the direction of lines when subjected to extensive deformation.
>
> - Regarding the challenge of dealing with overly smooth clothing regions, this is indeed a rather tricky aspect of virtual try-on tasks. Through preliminary experimental validation, we observed that **the perceptual loss in the overall loss function directly affects the smoothness of the clothing.** When the smoothness is excessive, there are fewer wrinkles in the clothing; conversely, there are more wrinkles when it's less smooth.
>
> - To validate this hypothesis, we conducted ablation experiments regarding the perceptual loss and $L_1$ loss. We controlled the hyperparameters of both to adjust their significance in the overall loss. The results of the ablation experiments are presented in Table 1 and Figure 8.2 (https://github.com/anony-conf/results-USC-PFN).
> ---
>
> |$\lambda_1$|$\lambda_p$|FID ↓|
> |:--:|:--:|:--:|
> |0|0|10.66|
> |1|0|17.45|
> |0|1|10.76|
> |1|1|**10.57**|
> |1|10|11.98|
> |10|1|11.46|
>
> ---
> - It can be observed that the network converges well only when the perceptual loss is present. When $\lambda_1=1$ and $\lambda_p=1$, meaning both are not artificially controlled, the convergence is optimal. Furthermore, from the fourth column of Figure 8.2, it can be seen that the wrinkles on the clothes are most realistic when both losses are utilized, while the second column without the perceptual loss exhibits an excessively smooth appearance. Thus, we have preliminary evidence that overly smooth clothing is a result of the perceptual loss. We can infer that the introduction of the adversarial loss would lead to more realistic wrinkles in the clothing.
>
> - However, overall, the appearance of wrinkles results in localized darkened pixels. If not controlled properly, these darkened regions might appear in unexpected areas of clothing, thereby obscuring the original details of the clothing. As a result, current virtual try-on methods primarily focus on the authenticity of clothing deformation first, and then address finer details like wrinkles.
> ---
> > We sincerely thanks for your dedicated efforts. We hope that our responses have provided you with new insights into our work. We look forward to you giving us a chance by increasing your approval of our paper. If you have more questions or encounter broken links, please comment, and we'll assist you quickly.

---

### Official Review · Reviewer_nwZk · 2023-07-27

**Soundness:** 3 good
**Presentation:** 2 fair
**Contribution:** 2 fair
**Rating:** 6
**Confidence:** 3

**Summary:**

The authors present a virtual garment try-on method. A cycle-consistency loss allows for self-supervision, i.e. the method does not require paired data (of the same person wearing different garments) as supervision. Unlike a previous method [2] that also uses cycle consistency, the same network weights are used for both the forward and backward step of the cycle. Additionally, a new garment deformation model based on Markov Random Fields is used instead of previous methods that use one of thin plate splines, appearance flow, or moving least squares. The authors show improved results over almost all previous methods, and similar results to one previous method [6].

**Strengths:**

- Cycle consistency using shared weights makes sense. It might seem like a small change, but does seem to require support from newly introduced losses.
- The new deformation model is a contribution, although it is unfortunately not evaluated separately from the full try-on model.
- The evaluation is reasonable and results show a significant advantage over the previous cycle-consistent method, and a significant advantage over most other methods (except one).

**Weaknesses:**

- The exact differences to [2] that give the proposed method an advantage over [2] are unclear. The authors allude to  advantages over [2] in several passages, but never fully describe what the differences to [2] are exactly (apart from re-using the same weights for both directions in the cycle) or where the advantages come from. For example:
	- In Section 2, the authors mention that [2] prioritizes paired garment-person images which can lead to difficulties with unpaired data. However, the proposed method also uses paired (garment, person) images, so what exactly is the difference? This needs to be clarified.
	- in Section 3.1, the authors mention a 'deconstruction of the human body' that leads to some problems in [2]. It is unclear what the authors refer to here. Do the authors refer to the Densepose descriptor used in [2]? If so, the authors would first have to mention that [2] needs to use the Densepose descriptor, so the reader has enough context to understand this statement. Just looking at the Eq 5 and 6, the only difference between the proposed method and [2] seems to be that the same weights are re-used for both directions. If there are any other differences, these need to be described explicitly near Eq 5 before referring to them. This needs to be clarified.
	- Also in Section 3.1, the authors mention that previous methods (including [2]) need an 'a-priori label'. Again, the authors need to describe what this refers to, otherwise the differences to [2] are hard to understand.
- The existing method [6] seems to have a similar performance. A discussion of other advantages over [6] would be good.
	- FID is better for [6] while SSIM is better for the proposed method.
	- PF-AFN look a bit more natural to me in all examples of Figure 4 except the first row, and maybe some details in the fourth row. It looks more natural to me because it seems to better follow the shape of the body, the proposed method looks flatter, as if it does not fully follow the body shape.
- The evaluation could be improved with additional ablations:
	- Using SIG with the same deformation method that [2] uses, to have a more direct comparison of using cycle consistency with shared vs. non-shared weights in the forward/backward steps.
	- Comparing the performance of NGD to previous deformation methods separately from the full try-on pipeline.
- The exposition is often confusing and should be improved.
	- The introduction emphasizes that the method only uses unpaired data, but it seems paired (garment A, person wearing A) data was still used. The introduction should clarify which kinds of pairs were not used as training. I assume the author refer to (person wearing A, person wearing B) pairs, but this needs to be clarified in the introduction.
	- The description of the deformation fields in Eq. 1 and the paragraph before it is a bit confusing. $\hat{f}$ is described as the optimal deformation field, but in the equation it is used as the variable that is being optimized (I would have expected  $\hat{f}$ to be the result of the minimization in Eq. 1, rather than the variable being minimized over).
	- At the end of Section 3, 'infinitely close' is probably not the right phrasing, since i) its unclear what 'infinitely close' means (why not say 'the same'?) and ii) the goal is to reconstruct $p$ with $\hat{p}$, but in practice, they will not be the same.
	- $R_{sm}$ is not defined in Eq. 2. The type of smoothing term should at least be described shortly.
	- A bit more information should be given in the weights $w_\phi$ in Eq. 2. How is the Gaussian constructed? How are its mean and variance computed? I assume something like the centroid and variance of pixel coordinates in the cloth region. A brief mention would be good.
	- In Eq. 2 and 3 it would be good to denote which variables the minimization is over (i.e. $\text{argmi}n_{\hat{f}}$ or $\text{argmi}n_{\phi}$.
	- In Eq. 3, it should probably be "$\dots\text{with} \dots$" instead of "$\dots s.t. \dots$"
	- Above Eq. 7 the authors mention that additional supervision is introduced for the upper body, but do not mention how it is used, what it consists of, etc. If this is described later on, it would be better to not mention this yet, to avoid confusion, or to re-structure the sections so the reader does not need to know information that will only be provided later on to understand the current text.
	- In Eq. 7 the notation is confusing. I did not follow why every variable has an additional ' in the notation. Why not use the same input notation as before: $(g', p)$, and output is $\hat{g}'$?
	- In Section 3.2, some design choices should be discussed:
		- Why is $\mathcal{G}_\mathcal{A}$ not used directly instead of $\mathcal{G}_\theta$? (I assume because the authors want to avoid using Densepose in the cycle training, but it should be discussed why, and ideally an ablation should be given.)
		- Why does $\mathcal{G}_\theta$ need to be pre-trained instead of directly starting training with $\mathcal{G}_\mathcal{A}$ and then finetuning?
	- Eq. 8 is unclear, it would be clearer to explicitly describe the loss used to train $\mathcal{G}_\theta$ in the second phase, probably something like $\min_\theta \mathcal{G}_\theta(g', p) - \mathcal{G}_\mathcal{A}(g', p_d, p_h)$
	- In Section 3.3, the definition of $L_{sec}$ should be given, or at least a short description of what this loss does.
	- In Eq. 9, $\mathcal{L}_{per}$ is not defined, it is only defined later on in the next subsection.
	- Above Eq. 15, step 1 and step 2 are not defined. These probably refer to the forward and backward steps in the cycle, but they should be defined more explicitly.
	- The loss in Eq. 15 needs a bit more motivation. why is backpropagation disabled if $\hat{p}'$ is more similar to $p$ than $\hat{p}$? Could this not happen quite frequently at the start of the training? Some discussion is needed.
	- A bit more information is needed how SSIM is computed. Is it computed by using the garment as input that the input person is already wearing, and is a held-out test set of paired data used?
	- In the ablation study when removing the MRF module, what deformation method is used instead?

Details:
- Near line 80: self-consistency consistency -> self-cycle consistency
- In Section 3.2, the paragraph titles should give the acronyms of the two modules in paranthesis (NGS, SIG), since the acronyms are used later on in the text.

**Questions:**

- Please clarify the setup that was used to compute SSIM.
- Please clarify all differences that [2] that could cause the improved performance shown in the results.
- Please describe all advantages of the proposed method over [6], and what you consider to be the most important advantages.

**Limitations:**

Some limitations have been discussed. Impacts from potentially biased datasets could be mentioned.

---

> ### Author Rebuttal · Authors · 2023-08-10
>
> > Thanks for your diligent efforts and valuable suggestions. We have provided both quantitative and qualitative results of our garment deformer on VITON and a high-definition VITON-HD (512×384) datasets (see https://github.com/anony-conf/results-USC-PFN).
> ---
> ### Weaknesses:
> - **W-A1:** The differences with [2] are: 1) [2] uses dual-network but only one network for inference, leading to convergence challenges and increased computational overhead. Our method achieves efficient convergence with a single shared-weight network, requiring fewer parameters and FLOPs. 2) [2] relies on human parsing as input during both training and inference, making the results sensitive to incorrect parsing. Our results only takes garment and person images as input, thus avoiding this issue. 3) [2] segments clothing and skin, which may cause artifacts at boundaries after their synthesis. 4) [2] introduces a multi-encoder network for feature extraction, adding complexity. Our method can utilize any generator as our training network, achieving results surpassing [2]. 5) [2] uses a globally deformable STN for clothing deformation. We introduce an MRF-based deformator achieving SOTA performance. 6) Our method has more advanced performance than [6]. In summary, our approach offers a novel solution for virtual try-on tasks.
> ---
> - **A1-1:** Sorry for your confusion. In Sec. 2, the "self-cycle training" is our own architecture, not [2]. [2] itself refers to its approach as 'disentangled cycle-consistency.'" Moreover, both our method and [2] utilize paired images.
> ---
> - **A1-2:** In Sec. 3.1, 'deconstruction of the human body' refers to [2] introduced a parsing  network to generate skin and clothing separately. Incorrect parsing can lead to erroneous try-on results. Our method only take clothing and person images as inputs, thus avoiding the issues.
> ---
> - **A1-3:** The term 'a-priori label' refers to the use of the Densepose or human parsing as input. Our network completely eliminates any such a-priori labels from the input.
> ---
> - **W-A2-1:** Thanks for your comments. FID is not universally suitable for [6], as there are some erroneous images in datasets, which might lead to slight disadvantages in FID. We utilized a publicly available augmented VITON dataset for the purpose of generalization validation. The table below demonstrates that our FID significantly outperforms PF-AFN.
>
> |Methods|SSIM ↑|FID ↓|
> |:--:|:--:|:--:|
> |ACGPN|0.81|20.75|
> |Cloth-flow|0.86|13.05|
> |PF-AFN|0.87|12.19|
> |**Ours**|**0.90**|**10.48**|
> ---
> - **W-A2-2:** In Fig. 4, PF-AFN exhibits excessive deformation of the lines on the abdomen and arms, which is highly unnatural and indicative of clothing deformation failure. Our results adhere to the body shape, especially in the arm regions. Some instances appearing flattened are due to our suppression of excessive folding during training, as an abundance of folds could potentially obscure clothing details.
> ---
> - **W-A3-1:** We replaced NGD with the STN from [2] to generate try-on results. The qualitative and quantitative results are shown in Table below and Fig. 8.1 (https://github.com/anony-conf/results-USC-PFN).
>
> |Methods|SSIM ↑|FID ↓|KID ↓|
> |:--:|:--:|:--:|:--:|
> |DCTON [2]|0.83|16.32|0.915|
> |**Ours**|**0.89**|**10.29**|**0.229**|
> ---
> - **W-A3-2:** We have conducted qualitative and quantitative experiments for NGD on both VITON (Table 1) and the high-definition VITON-HD datasets (Table 2). See Table below and Fig. 8.1.
>
> |Methods|Pub.|Warping|FID-P ↓|KID-P ↓|FID-UP ↓|KID-UP ↓|
> |:--:|:--:|:--:|:--:|:--:|:--:|:--:|
> ACGPN|CVPR 2020|TPS|42.10|2.009|41.48|2.048|
> DCTON|CVPR 2021|TPS|42.80|2.170|42.19|2.126|
> PFAFN|CVPR 2021|AF|22.81|0.785|23.90|0.860|
> SGAFN|CVPR 2022|AF|20.07|0.552|20.38|0.481|
> **Ours** (NGD)|/|**MRF**|**18.70**|**0.390**|**19.50**|**0.355**|
> ---
> |Methods|Pub.|Warping|FID-P ↓|KID-P ↓|FID-UP ↓|KID-UP ↓|
> |:--:|:--:|:--:|:--:|:--:|:--:|:--:|
> |VITON-HD|CVPR 2021| TPS|32.968|1.407|32.93|1.353|
> |HR-VITON|ECCV 2022|AF|25.499|0.926|24.826|0.759|
> |**Ours** (NGD)|/|**MRF**|**19.060**|**0.418**|**22.861**|**0.504**|
> ---
> - **W-A4:** Thanks for your professional and meticulous comments, which are crucial for enhancing the quality of our paper. We have made major revisions based on these suggestions. However, due to the limitations of the rebuttal space, we regret that we are unable to write down the answer to each of your concerns raised below individually. Once again, we sincerely appreciate your diligent efforts.
> ---
> ### Questions:
> - **Q-A1:** The setup that was used to compute SSIM has already been explained on the open-source webpage (https://github.com/anony-conf/USC-PFN/).
> ---
> - **Q-A2:** All differences between our work and [2] have been summarized in the aforementioned **W-A1**.
> ---
> - **Q-A3:** 1) [6] relies on prior knowledge to guide the student model, which may involve irresponsible teacher knowledge. Our approach does not employ a similar structure; instead, it self-guides for convergence. 2) [6] is unable to mitigate the impact of errors in the teacher knowledge, whereas we have avoided this aspect. 3) [6] designs a complex structured deformer, while we have the flexibility to adopt any generator.
> - The most significant advantage is that [6] requires meticulously designed and unique complex clothing deformer, whereas we can employ any network to serve the architecture, resulting in more realistic deformation effects. In try-on synthesis stage, [6] relies on a parser-based prior model, while we rely solely on our own approach. Hence, our method offers a completely novel approach and architecture for the virtual try-on task.
> ---
> ### Limitations:
> - **A:** We conducted experiments on VITON-HD and augmented VITON datasets to demonstrate the generalization and robustness of our method, unaffected by potential biased dataset influences.
> ---
> > We hope that our responses have provided you with new insights into our work. We look forward to you giving us a chance by increasing your approval of our paper.

---

> > ### Comment · Reviewer_nwZk · 2023-08-15
> > **Thanks for clarifications and additional experiments**
> >
> > Thanks for the thorough answers and clarifications! The additional experiments fill in some blind spots in the evaluation, especially showing a clearer advantage over [6] and showing the performance of SIG and NGD separate from each other. Assuming that the authors will add the clarifications and additional experiments to the final version of the paper, I will raise my score by one point.

---

### Author Rebuttal · Authors · 2023-08-10

> We sincerely appreciate the diligent efforts of the reviewers. We propose a novel parser-free self-cycle consistency framework, USC-PFN. To validate the effectiveness, robustness, and generalization of this architecture, we have added the following supplementary experiments:

1) Qualitative experiments of the clothing deformer NGD based on the VITON dataset.
2) Quantitative experiments of the clothing deformer NGD based on the VITON dataset.
3) Qualitative experiments of the clothing deformer NGD based on the high-definition **VITON-HD** dataset.
4) Quantitative experiments of the clothing deformer NGD based on the high-definition **VITON-HD** dataset.
5) Quantitative experiments of full network based on the augmented VITON dataset.
6) Qualitative experiments of full network on the high-definition **VITON-HD** dataset.
7) Quantitative experiments of full network on the high-definition **VITON-HD** dataset.
8) Computational complexity analysis of full network on the high-definition **VITON-HD** dataset.
9) Several other relevant ablation experiments.

- The tables of all the added quantitative experiments are presented below:
---
### Quantitative experiments of the clothing deformer NGD based on the VITON dataset.

|Methods|Publication|Warping|FID-P $\downarrow$|KID-P $\downarrow$|FID-UP $\downarrow$|KID-UP $\downarrow$|
|:--:|:--:|:--:|:--:|:--:|:--:|:--:|
CP-VTON [3]|ECCV 2018|TPS|43.95|2.233|42.13|2.112|
ACGPN [4]|CVPR 2020|TPS|42.10|2.009|41.48|2.048|
DCTON [2]|CVPR 2021|TPS|42.80|2.170|42.19|2.126|
PFAFN [6]|CVPR 2021|AF|22.81|0.785|23.90|0.860|
SGAFN [7]|CVPR 2022|AF|20.07|0.552|20.38|0.481|
**Ours** (MRF)|**This Work**|**MRF**|**18.70**|**0.390**|**19.50**|**0.355**|
---
### Quantitative experiments of the clothing deformer NGD based on the high-definition VITON-HD dataset.

|Methods|Publication|Warping|FID-P $\downarrow$|KID-P $\downarrow$|FID-UP $\downarrow$|KID-UP $\downarrow$|
|:--:|:--:|:--:|:--:|:--:|:--:|:--:|
|VITON-HD|CVPR 2021| TPS|32.968|1.407|32.93|1.353|
|HR-VITON|ECCV 2022|AF|25.499|0.926|24.826|0.759|
|**Ours** (MRF)|**This Work**|**MRF**|**19.060**|**0.418**|**22.861**|**0.504**|
---
### Quantitative experiments of full network based on the augmented VITON dataset.

|Methods|SSIM $\uparrow$|FID $\downarrow$|
|:--:|:--:|:--:|
|ACGPN|0.81|20.75|
|Cloth-flow|0.86|13.05|
|PF-AFN|0.87|12.19|
|**Ours**|**0.90**|**10.46**|
---
### Quantitative experiments of full network on the high-definition VITON-HD dataset.

|Methods|Publication|Parsing as input|SSIM $\uparrow$|FID $\downarrow$|KID $\downarrow$|
|:--:|:--:|:--:|:--:|:--:|:--:|
|CP-VTON|ECCV 2018|Y|0.791|30.25|4.012|
|ACGPN|CVPR2020|Y|0.858|14.43|0.587|
|DCTON|CVPR2021|Y|0.810|15.55|/|
|VITON-HD|CVPR2021|Y|0.843|11.64|0.300|
|HR-VITON|ECCV2022|Y|0.878|9.90|0.188|
|**Ours**|**This Work**|**N**|**0.899**|**9.10**|**0.159**|
---
### Computational complexity analysis of full network  on the high-definition VITON-HD dataset.

|Methods|Weight Size|#Params|FLOPs|FPS|
|:--:|:--:|:--:|:--:|:--:|
|VITON-HD|588M|154M|1689.7G|3.76|
|HR-VITON|586M|148M|1555.4G|4.09|
|**Ours**|**334M**|**87M**|**467.9G**|**22.19**|
---
### Computational complexity analysis of full network on the VITON dataset.

|Methods|#Params|FLOPs|FPS|FID-Clothing $\downarrow$|SSIM-TryOn $\uparrow$|
|:--:|:--:|:--:|:--:|:--:|:--:|
|ACGPN [4]|139M|206G|10|42.10|0.84|
|DCTON [2]|153M|194G|19|42.80|0.83|
|PF-AFN [6]|99M|69G|34|22.81|0.89|
|**Ours** (All ngf=64)|140M|46G|39|18.70|0.91|
|**Ours** (NGD ngf=32)|**87M**|**29G**|**40**|**18.85**|**0.91**|
---

> The remaining figures and tables for qualitative results, ablation experiments, and rebuttals related to the experiments are all included in the following link and attachment.
---
- https://github.com/anony-conf/results-USC-PFN

---

- https://github.com/anony-conf/USC-PFN/
- (The checkpoints will be made publicly available immediately.)

---
---
> We sincerely thank you for your dedicated efforts. We hope that our responses have provided you with new insights into our work. We look forward to you giving us a chance by increasing your approval of our paper. If you have more questions or encounter broken links, please comment, and we'll assist you quickly.

---

### Decision · Program_Chairs · 2023-09-21

**Decision:**

Accept (poster)

**Comment:**

Four reviewers favor acceptance, while one is borderline. The reviewer recommending rejection (Reviewer RUsU) was initially concerned about the presentation of the MRF and its novelty over previous methods based on appearance flow. After the discussion, they suggested that these issues had mostly been addressed, but felt that the discussion of the MRF was not sufficiently detailed in the current version. The AC agrees that the paper has clarity issues (as well as a number of minor formatting issues, such as inconsistently capitalized section titles and writing issues. On balance, the AC recommends acceptance, but urges the authors to improve the clarity of the paper in a revision.